# A proton-gated channel identified in the centipede antenna

Wenqi Dong [ID] [1,2,3,6], Licheng Yuan [ID] [1,2,3,6], Jiangming Shang [ID] [4,5,6], Fan Yang [ID] [4,5], Shilong Yang [ID] [1,2,3], Xiancui Lu [ID] [1,2,3], Qian Wang[1,2,3], Anna Luo[1,2,3], Jiheng Geng[1,2,3], Jiatong Cheng[1,2,3], Runze Li[1] & Yunfei Wang [ID] [1,2,3] ✉

## Abstract

Acid sensing is essential for various biological processes in animals, yet it exhibits species-specific characteristics. In this study, we identified a proton-dissociation-permeated sodium channel (PDPNaC1) in the antennal sensory neurons of the centipede *Scolopendra subspinipes mutilans*. PDPNaC1, which is permeable to monovalent cations, assembles as a homotrimer. Unlike most proton-gated channels, where proton binding induces currents, PDPNaC1's transient ion-permeable state is triggered by proton dissociation. By resolving the high-resolution cryo-electron microscopy (cryo-EM) structure of PDPNaC1, combined with mutagenesis and electrophysiological analyses, we identified Gly378, rather than the Gly-Ala-Ser tract, as a key determinant of ion selectivity. Furthermore, Ser376, located in the ion-permeable pathway, likely serves as a proton-binding site, leading to an $H^+$-blocking effect that results in proton-dissociated currents. Thus, the identification of PDPNaC1 suggests the remarkable diversity of proton responses and molecular mechanisms in DEG/ENaC family.

Keywords Acid-sensing; Antennal Sensory Neurons; Cryo-electron Microscopy; Proton-gated Channel; PDPNaC1
Subject Categories Membranes & Trafficking; Neuroscience; Structural Biology

## Introduction

Animals have evolved diverse molecular strategies to detect external and internal pH variations, primarily through proton sensitivity exhibited by G-protein-coupled receptors (GPCRs) and ion channels. Certain GPCRs, such as GPR4, GPR31, GPR65, GPR68, GPR132, and GPR151, are activated by elevated extracellular proton concentrations, triggering a cascade of signal transduction events that regulate cellular processes like ion transport, secretion, and apoptosis (Holzer, 2011; Yang et al, 2019). Beyond GPCRs, proton-gated channels also play a pivotal role in pH sensing. These channels include acid-sensing ion channels (ASICs), transient receptor potential (TRP) ion channels (e.g., TRPV1, TRPV4), otopetrin channels (OTOPs), proton-activated chloride (PAC) channels, two-pore-domain potassium (K2P) channels, ionotropic purinoceptors (P2X), and the transmembrane protein 175 (TMEM175) in lysosome (Caterina et al, 1997; Holzer, 2009; Hu et al, 2022; Suzuki et al, 2003; Tu et al, 2018; Waldmann et al, 1997; Yang et al, 2019). Each of these channels exhibits distinct pH sensitivities, ion selectivity, and channel kinetics, enabling the modulation of membrane potential, cell excitability, intracellular acidity, cell volume, and organelle function.

While proton-sensitive receptors play an evolutionarily conserved role in acid sensing, the specific functional demands for detecting acidic conditions have diversified across species. For instance, mosquitoes are attracted to acidic volatile compounds in human sweat, which guide them to hosts for blood-feeding (Raji et al, 2019). In fruit flies, acidic compounds produced during fruit fermentation stimulate egg-laying behavior (Carpenter and Broadbent, 2009; Joseph et al, 2009). Soil-dwelling myriapods, equipped with specialized antennal sensilla (Sombke et al, 2011), exhibit remarkable sensitivity to habitat pH (Morgan, 2011, Yang et al, 2025). These diverse demands for acid detection across species may have driven the evolution of species-specific proton-sensitive receptors, which enable adaptation to distinct ecological niches and physiological needs.

In this study, we performed single-nucleus RNA sequencing to characterize the neuronal populations in the antennae of the centipede (*Scolopendra subspinipes mutilans*). Within a cluster of primary sensory neurons beneath the antennal surface, we identified a member of the DEG/ENaC family that forms a proton-gated ion channel. Interestingly, this channel elicits robust inward currents upon proton dissociation and is therefore termed the proton-dissociation-permeated sodium channel (PDPNaC1). Guided by the structural analysis of PDPNaC1, we functionally identified the ion-selective filter and investigated the mechanism of the $H^+$-blocking effect. Thus, the identification of PDPNaC1 underscores the diversity of acid-sensing both at the molecular and species levels.

[1]College of Wildlife and Protected Area, Northeast Forestry University, Harbin, Heilongjiang, China. [2]Key Laboratory of National Forestry and Grassland Administration on Wildlife Protection, Harbin, Heilongjiang, China. [3]Heilongjiang Key Laboratory of Complex Traits and Protein Machines in Organisms, Harbin, Heilongjiang, China. [4]Department of Biophysics and Disease Center of the First Affiliated Hospital, Zhejiang University School of Medicine, Hangzhou, Zhejiang, China. [5]Liangzhu Laboratory, Zhejiang University Medical Center, Hangzhou, Zhejiang, China. [6]These authors contributed equally: Wenqi Dong, Licheng Yuan, Jiangming Shang. ✉E-mail: wangyunfei@nefu.edu.cn

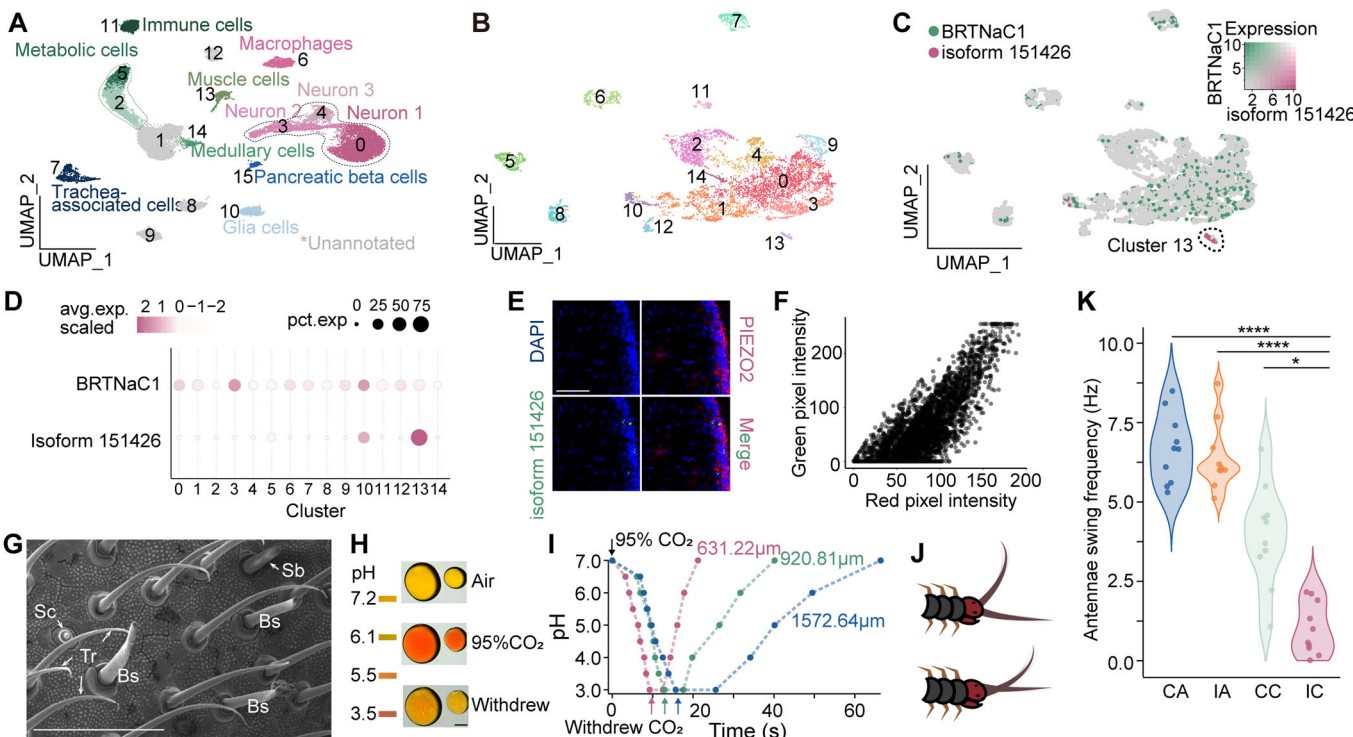

**Figure 1. The snRNA-seq analysis, tissue distribution, and animal behavior.**

(A) UMAP plot of the 16 cell clusters generated from 21,837 nuclei obtained from the antennae of the centipede *S. s. mutilans*. (B) Reclustering and UMAP visualization of 7534 neuronal cells. (C) Feature plots on UMAP visualization showing *isoform 151426* and *BRTNaC1* expression in antennal neuronal cell subpopulations. (D) The dot plot shows the expression of *isoform 151426* and *BRTNaC1* across 15 subclusters of centipede antennal neurons (re-clustered as shown in **B**). Dot size indicates the percentage of cells expressing the gene, and color intensity reflects the average expression level within each cluster. (E) Representative immunostaining images of isoform 151426 and PIEZO2 in the centipede antenna. Scale bar: 50 μm. (F) Scatter plot showing PIEZO2 (red) and PDPNaC1 (green) signal correlation in the antennal section (E). Pearson's correlation (PCC = 0.81), Manders' coefficients (M1 = 0.749, M2 = 1.000), and non-parametric correlations (ρ = 0.765, τ = 0.608) all support strong colocalization (Costes P = 1.00). (G) SEM structure of antenna surface cuticle. Tr: trichome; Bs: beak-like sensilla; Sb: sensilla brachyconica; Sc: sensory cone. Scale bar: 30 μm. (H) Schematic diagram of the color change of methyl red droplets when air or 95% $CO_2$ was applied and withdrawn. Scale bar: 500 μm. (I) pH changes over time of methyl red droplets with different diameters during continuous 95% $CO_2$ exposure and after $CO_2$ removal. The black arrow represents the application of $CO_2$, and the colored arrows indicate the time point of $CO_2$ withdrawal. (J) Schematic diagram of centipede antenna swinging. (K) Swinging frequency of centipede antennae under different environments ($n = 10$ centipedes for each group). CA: Continuous Air; IA: Intermittent Air; CC: Continuous $CO_2$; IC: Intermittent $CO_2$. Data information: In (**K**), Kruskal–Wallis rank-sum test with post hoc Dunn test was used for pairwise comparison, $p_{(CC)} = 0.0249$, $p_{(IA)} = 6.6 \times 10^{-6}$, $p_{(CA)} = 1.0 \times 10^{-6}$. All individual data points are shown. Source data are available online for this figure.

# Results and discussion

## A DEG/ENaC member expressed in antennal receptor cells

We performed single-nucleus RNA sequencing (snRNA-seq) on the antennae using the 10x Genomics platform. After filtering out low-quality nuclei, 21,837 nuclei were retained for further analysis, with a median of 882 genes and 1338 transcripts per nucleus. Following dimensionality reduction and unsupervised clustering, the nuclei were grouped into 16 clusters based on gene expression patterns (Fig. 1A). Twelve of these clusters were annotated based on the expression of defining marker genes (Appendix Table S1). Within clusters 0, 3, and 4, identified as neurons, we conducted a reclustering analysis on a dataset of 7534 nuclei (Fig. 1B) and analyzed the differential gene expression within these subclusters (Appendix Fig. S1A). We identified a previously unannotated gene (*isoform 151426*) exhibiting subcluster 13-specific expression patterns (Appendix Fig. S1A). This gene comprises nine exons

and eight introns, encoding a 421-amino acid protein (Appendix Fig. S1B). Phylogenetic analysis demonstrates that isoform 151426 belongs to the DEG/ENaC (Degenerin/Epithelial Sodium Channel) family (Appendix Fig. S1C and Appendix Table S2). The protein shares 23% amino acid sequence similarity to its closest relative BRTNaC1 (Appendix Fig. S1D), a thermal receptor found in the same species (Chen et al, 2025; Yao et al, 2023). Unlike *BRTNaC1*, which is ubiquitously expressed across multiple neuronal subclusters, *isoform 151426* expression is restricted to a specific group of receptor cells that express several gene markers, including *PIEZOs*, *TRPM3*, *CHAT*, *GRIK4*, and *NETO2* (Fig. 1C,D; Appendix Fig. S2). Immunofluorescent staining revealed that the fluorescence signals corresponding to isoform 151426 protein localized beneath the antennal surface, demonstrating colocalization with PIEZO2, the primary mechanosensory receptor (Fig. 1E,F).

The antennal surface of centipedes is adorned with various types of sensilla (Fig. 1G), which are small cuticular structures likely filled with lymph-like body fluid. These microstructures may undergo rapid chemical changes in response to environmental stimuli, such

as pH fluctuations induced by ambient $CO_2$. To emulate this process, we conducted experiments using droplets of varying diameters. Upon application and removal of $CO_2$, we observed rapid acidification and pH recovery within seconds. Notably, the pH transition rate was inversely correlated with droplet size (Fig. 1H,I; Movie EV1), with the transition for a 1-μm droplet predicted to reach the microsecond level (Appendix Fig. S3). Interestingly, the centipede's antenna exhibited greater sensitivity to $CO_2$ fluctuations than to persistent $CO_2$ application (Fig. 1J,K; Movie EV2). DEG/ENaC channels are frequently sensitive to extracellular pH, prompting the hypothesis that isoform 151426 may function as an acid-sensitive receptor. In addition, this isoform is selectively expressed in antennal sensory neurons located at the surface of the antenna, a position well-suited for direct exposure to ambient chemical cues. Taken together, these features suggest that neurons expressing isoform 151426 may act as primary sensors of environmental pH, potentially responding to acidic stimuli such as $CO_2$ exposure. These findings prompted further investigation into the electrophysiological properties of isoform 151426.

## A proton-gated monovalent cation channel

Whole-cell recordings of PAC knock-out HEK293T cells expressing isoform 151426 revealed that this protein forms proton-gated channels on the plasma membrane. Upon exposure to low pH, these cells exhibited small but sustained currents, whereas proton dissociation triggered transient yet large inward currents (Fig. 2A). Based on its temporal response to changes in $H^+$ concentration, we termed this channel the proton-dissociation-permeated sodium channel (PDPNaC1). To simplify the analysis of the proton response, $H^+$-induced PDPNaC1 currents were classified into two components: $I_1$, representing the small currents induced by proton binding, and $I_2$, representing the large currents triggered by proton dissociation (Fig. 2A, right). The concentration-response fitting to the Hill equation indicated the half-maximal effective concentration ($EC_{50}$) at pH ~3.60 for $I_1$ and at pH ~3.61 for $I_2$, suggesting their comparable pH dependence (Fig. 2B). Like other DEG/ENaC members, PDPNaC1 was potently inhibited by zinc, with a half-maximal inhibitory concentration ($IC_{50}$) of $22.01 \pm 3.6$ μM (Fig. 2C,D). We also found that calcium ions attenuate the pH dependence of PDPNaC1, decreasing the $pH_{50}$ to 3.50 ($I_1$) and 3.57 ($I_2$) in the presence of 10 μM calcium (Appendix Fig. S4). This mode of regulation, consistent with that in ASIC channels (Immke and McCleskey, 2003; Molton et al, 2024; Roy et al, 2022), suggests that calcium-dependent modulation of proton sensitivity may represent a conserved mechanism within the DEG/ENaC family.

Peak current amplitudes at $-80$ mV were used as a qualitative indicator to assess whether PDPNaC1 conducts various cations. Under symmetric 140 mM NMDG-Cl conditions, a robust inward current was observed at pH 3.0 (Fig. 2E). However, no current was detected when the extracellular pH was shifted from 3.0 to 7.0 (Fig. 2E), indicating that the channel is impermeable to $NMDG^+$ and $Cl^-$, and that the inward current observed at low pH 3.0 is primarily carried by protons. Further testing with various cation substitutions revealed that both $I_1$ and $I_2$ currents are permeable to the same set of monovalent cations ($H^+$, $Na^+$, $K^+$, and $Cs^+$), but not to divalent cations such as $Mg^{2+}$ (Fig. 2E–H). To more accurately

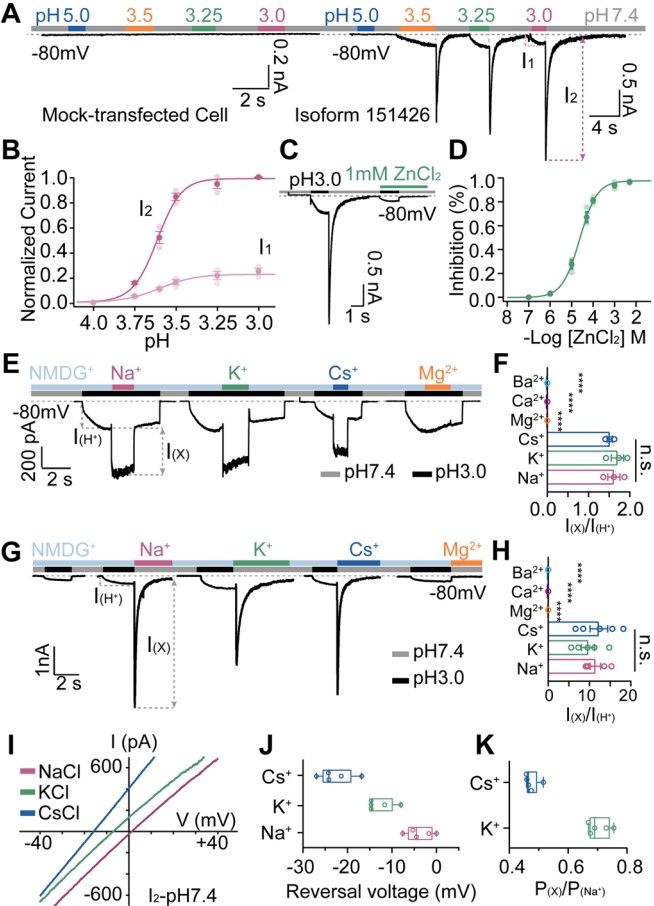

Figure 2. Proton-induced response of isoform 151426.

(A) Representative current traces of Mock-transfected (left) and isoform 151426-expressing (right) cells under different pH conditions ($V_m = -80$ mV). (B) Dose–response curves of isoform 151426 activated by proton ($n = 3$ cells for each group). Data were fitted using the Hill equation. (C) Representative proton binding and dissociation-induced current of PDPNaC1 at pH 3.0 in the absence and presence of 1 mM $ZnCl_2$ ($V_m = -80$ mV). (D) Dose–response curves of PDPNaC1 inhibition by $ZnCl_2$ ($n = 3$ cells for each group). Data were fitted using the Hill equation. (E) PDPNaC1 currents were evoked by pH 3.0 solutions containing 140 mM $Na^+$, $K^+$, or $Cs^+$ or 110 mM $Mg^{2+}$, replacing 140 mM $NMDG^+$ in the extracellular solutions as indicated ($V_m = -80$ mV). (F) The ratio of $I_{(X)}$ to $I_{(H^+)}$ corresponds to (E) ($n = 5$ cells for each group). $p_{(Na\ vs\ K)} = 0.9805$, $p_{(Na\ vs\ Cs)} = 0.4649$, $p_{(Na\ vs\ Ca/Mg/Ba)} = 8.2 \times 10^{-5}$. (G) PDPNaC1 currents were evoked by pH 3.0 solutions containing 140 mM $NMDG^+$, and by switching the extracellular solution from pH 3.0 to 7.4, with 140 mM $Na^+$, $K^+$, or $Cs^+$ or 110 mM $Mg^{2+}$ ($V_m = -80$ mV). (H) The ratio of $I_{(X)}$ to $I_{(H^+)}$ corresponds to (G) ($n = 5$ cells for each group). $p_{(Na\ vs\ K)} = 0.1642$, $p_{(Na\ vs\ Cs)} = 0.6912$, $p_{(Na\ vs\ Ca/Mg/Ba)} = 7.4 \times 10^{-8}$. (I) Representative I–V traces for PDPNaC1-expressing cells with 140 mM NaCl in the pipette and equimolar extracellular cations. Currents were recorded during the transient current $I_2$ at pH 7.4 using voltage ramps from $-60$ mV to $+60$ mV over 120 ms. (J) Reversal potentials for PDPNaC1 in the presence of different cations ($n = 5$ cells for each group). (K) Relative ion permeabilities for PDPNaC1 ($n = 5$ cells for each group). Data information: In (F, H), data are presented as mean ± SEM (Unpaired two-tailed Student's t-test). In (J, K), data are presented as box-and-whisker plots showing median (center line), 25th and 75th percentiles (box limits), and minimum to maximum values (whiskers). All individual data points are shown. Source data are available online for this figure.

quantify the ion selectivity, we performed voltage ramp recordings on $I_2$ currents, which occur after proton removal and are therefore not confounded by the proton electrochemical gradient that affects $I_1$. These measurements revealed that PDPNaC1 exhibits a modest preference for $Na^+$ over $K^+$ and $Cs^+$ (Fig. 2I–K). This contrasts sharply with classic acid-sensitive receptor ASICs in the DEG/ENaC family, which exhibit strong selectivity for sodium ions and are nearly impermeable to cesium ions (Hanukoglu, 2017; Vallee et al, 2021) (Appendix Fig. S5). Therefore, a high-resolution structure may allow us to pinpoint the unique electrophysical properties of the PDPNaC1 channel.

## Three-dimensional structure of the PDPNaC1 channel

We overexpressed PDPNaC1 in HEK293S cells to obtain homogeneous samples for cryo-electron microscopy (cryo-EM). At pH 8.0, approximately 121,831 selected particles yielded an EM reconstruction at an overall resolution of 3.03 Å (Appendix Fig. S6). The architecture of PDPNaC1 is composed of three homologous subunits, each featuring two transmembrane (TM) helices (Fig. 3A,B). These helices are connected by a linker region that bridges the transmembrane and extracellular domains (Appendix Fig. S7). The N- and C-termini flanking the transmembrane regions of PDPNaC1 remain unresolved, suggesting that these regions are either highly flexible or not well-defined in the current structural context.

The extracellular domain of each PDPNaC1 subunit, containing 12 β-strands and 6 α-helices, forms the palm, thumb, finger, and β-ball regions (Appendix Fig. S7), which are common features among DEG/ENaC family members (Jasti et al, 2007; Noreng et al, 2018; Yoder et al, 2018). Notably, despite sharing these conserved structural features, PDPNaC1 lacks the extracellular knuckle domain (Appendix Fig. S7), which mediates proton-induced activation and desensitization rearrangements in cASIC1a (Gwiazda et al, 2015). The absence of this structural domain is likely responsible for the lack of fast inactivation in the PDPNaC1 channel. To identify the potential proton-sensor, we calculated the protein charge distribution of PDPNaC1 (at pH 8.0) and identified a negatively charged cavity near the extracellular domain apex, formed by residues D227, D145, and Y143 in the β-ball and palm domains (Appendix Fig. S8). This differs from cASIC1a, in which the acidic pocket formed by the α5 helix of the thumb domain and the acidic loop of the finger domain has been implicated as the primary proton sensor (Korkosh and Tikhonov, 2023; Sun et al, 2018). This negatively charged pocket in PDPNaC1 may function as a proton-sensing module, although further experimental validation is required.

Within the transmembrane domain, a solvent-accessible pathway formed by the TM2 helices indicates the location of the pore region (Fig. 3C). The pore radius profile corresponds to the non-conductive state observed in PDPNaC1 at pH 8.0. Along the ion pathway, we observed two constriction sites corresponding to residues Asp372 and Gly375 with pore radii of 0.86 Å and 0.39 Å, respectively (Fig. 3D). These distances are insufficient to accommodate a sodium ion (radius 1.02 Å), thereby constituting two gates (upper and lower) within the central pathway, analogous to those observed at Asp433 and Gly436 in the cASIC1a channel (Yoder et al, 2018).

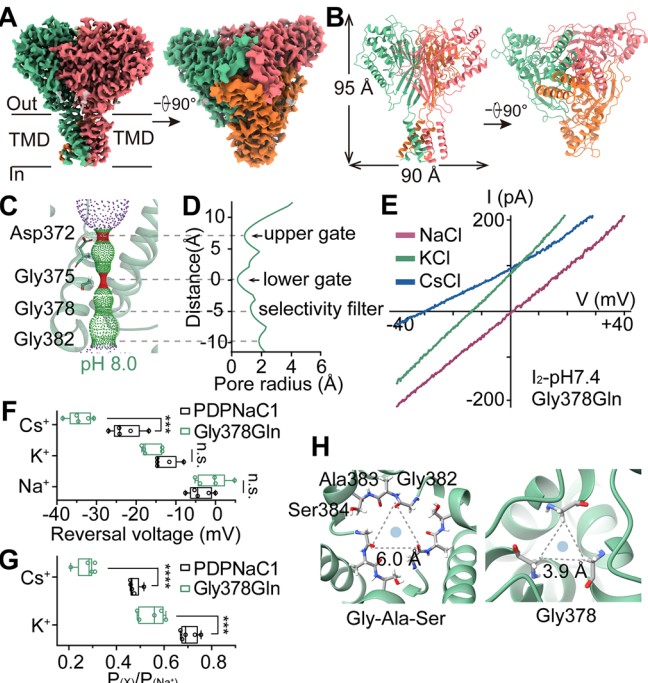

**Figure 3. Overall structure and ion-selectivity of PDPNaC1.**

(A) Side view (left) and bottom view (right) of the cryo-EM map of PDPNaC1, with each subunit individually colored. The black bars indicate the position of the cell membrane. (B) Cartoon representation of PDPNaC1 in side view (left) and bottom view (right), with each subunit individually colored (PDB: 9JF7). (C, D) Pore radius profiles for PDPNaC1 calculated using HOLE software (pore radius: red < 1.15 Å < green < 2.3 Å < purple) (Smart et al, 1993). (E) Representative I–V relationships for Gly378Gln in the presence of different cations with 140 mM NaCl in the pipette. Currents were recorded during the transient current $I_2$ at pH 7.4 using voltage ramps from −60 to +60 mV over 120 ms. (F) Reversal potentials for PDPNaC1-expressing and Gly378Gln-expressing cells in the presence of different cations ($n = 5$ cells for each group). $p$ for Na: 0.2206; $p$ for K: 0.0815; $p$ for Cs: 0.0008. (G) Relative ion permeabilities for PDPNaC1 and Gly378Gln ($n = 5$ cells for each group). $p$ for K: 0.0007; $p$ for Cs: $= 2.3 \times 10^{-6}$. (H) View of the Gly-Ala-Ser motif and Gly378 of PDPNaC1 from the intracellular side, showing the distance between Gly382 (6.0 Å) and Gly378 (3.9 Å), respectively. The blue circles denote the permeable pathway. Data information: In (F, G), data are presented as box-and-whisker plots showing median (center line), 25th and 75th percentiles (box limits), and minimum to maximum values (whiskers). All individual data points are shown (unpaired two-tailed Student's t-test). Source data are available online for this figure.

## Gly378 mutations enhance ion selectivity in PDPNaC1

A conserved Gly-Ala-Ser motif (Gly382, Ala383, Ser384), previously shown to stabilize the pore and thereby indirectly contribute to ion permeation in ASIC channels (Baconguis et al, 2014), is also present in PDPNaC1 (Appendix Fig. S9). We introduced single-point mutations to test whether this motif plays a role in the ion selectivity of PDPNaC1. Compared to the wild-type channel, none of the mutants exhibited altered ion selectivity (Appendix Fig. S10), suggesting that the Gly-Ala-Ser motif is unlikely to serve as the selectivity filter in PDPNaC1. Furthermore, in other DEG/ENaC channels, the GAS belt interacts with the 'His-Gly' (HG) motif to stabilize the pore architecture and indirectly

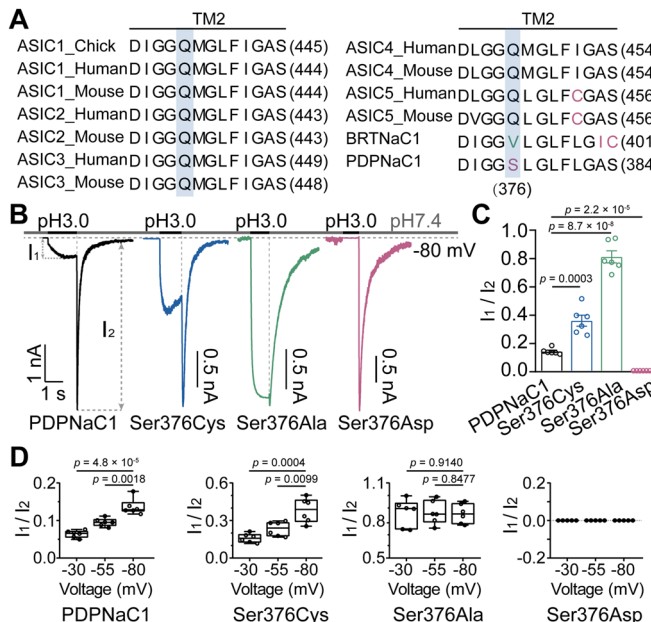

**Figure 4. Ser376 serves as the proton-binding site in PDPNaC1.**

(A) Partial sequence alignment of the TM2 helix in DEG/ENaC family members. (B) Representative current trace of PDPNaC1, Ser376Cys, Ser376Ala and Ser376Asp in response to proton binding and dissociation ($V_m = -80$ mV). (C) The ratio of $I_1$ to $I_2$ at pH 3.0 ($n = 6$ cells for each group). (D) The $I_1/I_2$ ratio of PDPNaC1 and its mutants was measured across different voltages ($n = 6$ cells per group). Data information: In (C), data are presented as mean ± SEM. In (D), data are presented as box-and-whisker plots showing median (center line), 25th and 75th percentiles (box limits), and minimum to maximum values (whiskers). All individual data points are shown (Unpaired two-tailed Student's t-test). Source data are available online for this figure.

facilitate ion permeation (Yoder and Gouaux, 2020). However, the PDPNaC1 channel lacks a canonical HG motif (Appendix Fig. S9), and this structural divergence may account for PDPNaC1's broader cation selectivity relative to ASIC channels.

Above the Gly-Ala-Ser motif, we identified a narrow constriction site where Gly378 from each subunit converges to form a hydrophobic seal, with a pore radius of 1.28 Å (Fig. 3D). Notably, substitution at position 378 (mutants Gly378Ser and Gly378Gln) significantly altered ion selectivity, reducing the permeability ratios of monovalent cations to $Na^+$, especially for $Cs^+$ (Fig. 3E–G; Appendix Fig. S10), indicating that the mutants enhance selectivity for $Na^+$. These results suggest that Gly378 contributes to maintaining a permissive pore constriction, thereby enabling broader ion permeability in the wild-type channel. Unlike other DEG/ENaC members that show a preference for sodium, PDPNaC1 may play a role not only in action potential generation but also in regulating the balance of monovalent cations to maintain ion homeostasis. The conserved Gly-Ala-Ser motif in ASICs and Gly378 in PDPNaC1 each contributes to cation selectivity (Fig. 3H), indicating that structural divergence in pore-lining residues underlies functional diversity of ion selectivity among DEG/ENaC family members.

## Proton inhibition of PDPNaC1 mediated by residue Ser376

The most intriguing distinction between PDPNaC1 and typical ASICs lies in the transition from $I_1$ to $I_2$. Given the similar ion

selectivity of $I_1$ and $I_2$, one possible explanation for this phenomenon is that protons drive pore opening while simultaneously inhibiting cation permeation, either by directly competing for cation-binding sites or through electrostatic interactions. When protons dissociate from the ion permeation pathway, the pore remains open, resulting in a transient permeative state to monovalent cations (Brelidze and Magleby, 2004; Lee and Zheng, 2015). To test this hypothesis, we conducted a sequence alignment of the TM2 helix among DEG/ENaC family members and found that Ser376 is a unique residue in PDPNaC1 compared to its evolutionary relatives (Fig. 4A; Appendix Fig. S9). Substituting this residue with cysteine increased the $I_1$ currents. Additionally, the alanine substitution resulted in $I_1$ currents with an amplitude comparable to that of the $I_2$ currents (Fig. 4B,C). Since $I_2$ could be interpreted as a transient ion-permeable state in the absence of protons, these results suggest that the proton dissociation from Ser376 likely mediates the transition from $I_1$ to $I_2$. If this is the case, the presence of a negatively charged amino acid at this position could enhance the $H^+$-blocking effect by increasing proton affinity, leading to a reduced $I_1$ and an unaffected $I_2$. Consistently, we found that the Ser376Asp mutant exhibited a non-conducting state during proton binding but produced large currents upon removal of the acidic solution (Fig. 4B,C). Moreover, we evaluated the voltage dependence of proton inhibition in wild-type PDPNaC1 and Ser376 mutants. The $I_1/I_2$ ratio increased with more negative membrane potentials in both the wild-type and Ser376Cys channels (Fig. 4D), indicating that proton block is progressively relieved under stronger hyperpolarizing conditions. In contrast, the Ser376Ala mutant showed no voltage dependence (Fig. 4D), consistent with the near-complete loss of proton affinity. Notably, in the Ser376Asp mutant, the $I_1$ component was abolished across all tested voltages (Fig. 4D), likely due to strong, constitutive proton binding and persistent channel block. These findings support the conclusion that proton block in PDPNaC1 occurs within the transmembrane electric field and involves a pore-localized inhibitory mechanism centered on Ser376.

## Conformational rearrangement at site 376 during channel opening

Given the buried conformation of Ser376 in the closed state of PDPNaC1 (Fig. 5A), we introduced the unnatural amino acid L-3-(6-acetylnaphthalen-2-ylamino)-2-aminopropanoic acid (ANAP) to evaluate the proton-induced conformational rearrangements. As a fluorescent probe, ANAP is highly sensitive to environmental polarity, showing a redshift in emission as polarity increases (Lee et al, 2009). We incorporated ANAP at Ser376 and Leu369 sites to test their conformational changes during channel gating (Fig. 5B,C; Appendix Fig. S11). The Ser376ANAP mutant exhibited a significant red shift in ANAP emission during both proton application and washout phases. In contrast, no change was observed in the emission peak of the Leu369ANAP mutant across varying pH conditions (Fig. 5D–F). This result suggests a conformational change in the Ser376 side chain from a buried to an exposed state. Moreover, free ANAP exhibited a ~10 nm blue shift when the pH was lowered from 7.0 to 3.0 (Appendix Fig. S12), indicating that the red shift observed for Ser376ANAP is not due to the intrinsic pH sensitivity of the fluorophore. Therefore, the local environmental change at Ser376 is likely even more pronounced

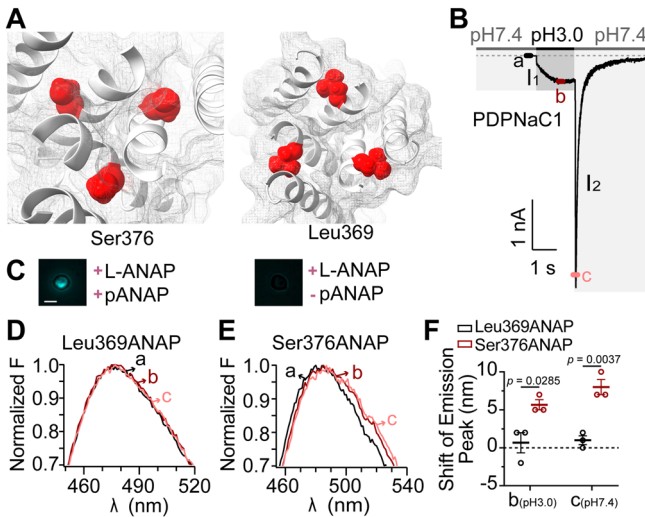

**Figure 5. Increased exposure of the Ser376 side chain upon PDPNaC1 opening.**

(A) Location of Ser376 and Leu369 on PDPNaC1 structure in apo state (pH 8.0), focus on the top views of sites Ser376 and Leu369. (B) Representative current trace of PDPNaC1 recorded under a constant holding potential of −80 mV. Points a, b, and c indicate time points at which fluorescence emission spectra were acquired. (C) Representative images of cells expressing ANAP-incorporated PDPNaC1 in the presence or absence of the pANAP vector. Pseudocolors for ANAP are used. Scale bar: 10 μm. (D, E) Representative emission spectra of Leu369ANAP- and Ser376ANAP-expressing cells were recorded at the time points indicated in (B), corresponding to pH 7.4 (a), pH 3.0 (b), and the pH transition from 3.0 (b) to 7.4 (c). (F) Summary of ANAP emission shifts at Leu369 and Ser376 in response to proton application (b, pH 3.0) and removal (c, pH 7.4) (n = 3 cells for each group). Data information: In (F), data are presented as scatter plots showing mean ± SEM. All individual data points are shown (Unpaired two-tailed Student's t-test). Source data are available online for this figure.

than the observed fluorescence shift suggests. Together, these findings indicate that residue Ser376 is a critical site for proton binding in the open state of PDPNaC1, exerting a H+-blocking effect that inhibits the flow of monovalent cations along the ion pathway.

## The gating mechanism and evolution of PDPNaC1

Similar H+-blocking effects have been observed in many ion channels, including LTCC, CNG, and BK channels (Benitah et al, 1997; Brelidze and Magleby, 2004; Chen et al, 1996; Lee and Zheng, 2015; Root and MacKinnon, 1994). In L-type $Ca^{2+}$ and CNG channels, protons inhibit ion permeation by binding to negatively charged glutamate residues near the pore entrance (Chen et al, 1996; Root and MacKinnon, 1994). In BK channels, protons competitively interact with the binding sites of potassium ions (Brelidze and Magleby, 2004). In the case of PDPNaC1, we identified Ser376 as the proton-binding site, which leads to the inhibitory effect of protons at low pH (Fig. 4B,C). Interestingly, we observed a transition of Ser376 from a buried to an exposed state upon proton binding (Fig. 5E,F), suggesting that the interaction between the protons and Ser376 may be extremely rapid, effectively inhibiting ion permeation during pore opening. Given the high pKa

of the serine hydroxyl group, direct protonation is unlikely under physiological conditions. However, the hydroxyl oxygen of Ser376 may interact with protonated water clusters, such as Zundel ($H_5O_2^+$) or Eigen ($H_9O_4^+$) cations (Wu and Voth, 2003), providing a potential mechanism for proton recognition through hydrogen bonding. Since the Ser376Ala mutant exhibited robust activation upon proton binding (Fig. 4B), H+-induced pore opening and ion permeation inhibition appear to be independent events. Additionally, the large currents observed upon proton dissociation suggest that the unbinding kinetics between the proton and Ser376 are rapid, occurring faster than the conformational change from the open to closed state of PDPNaC1. The state-dependent orientation of the Ser376 side chain suggests that other proton-binding sites responsible for the H+-blocking effect in certain channels may reside in residues that undergo buried-exposed transition during channel gating.

The DEG/ENaC superfamily exhibits remarkable diversity in gating mechanisms and physiological roles across various animal species. For example, DEG/ENaC members contribute to pain sensation and sodium absorption in mammals (Canessa et al, 1994; Hung et al, 2023), touch sensation in nematodes (O'Hagan et al, 2005), thermosensation in centipedes (Yao et al, 2023), and neuropeptide sensitivity in mollusks and hydra (Assmann et al, 2014; Dandamudi et al, 2022). From an evolutionary perspective, our study demonstrates that the DEG/ENaC family can respond to a wide range of pH values. Based on its primary sensory neuronal expression and electrophysiological response to pH changes (Figs. 1E and 2A), we speculate that PDPNaC1 likely functions as a molecular sensor for detecting rapid pH fluctuations at physiological levels. The activation of PDPNaC1 may be facilitated by the high frequency of antennal movements in centipedes (Guizze et al, 2016), potentially enabling rapid proton binding and unbinding of the channel. In this scenario, repeated PDPNaC1 activation generates action potentials in these nociceptive neurons, which trigger warning signals that help centipedes avoid mammalian predators in harmful acidic environments. Consequently, the neuronal localization and gating kinetics of PDPNaC1 likely work in synergy with antennal movements to detect pH fluctuations—such as those caused by predator respiration—in natural environments. This may confer an adaptive advantage to centipedes by enhancing their ability to sense transient acidic conditions.

## Methods

**Reagents and tools table**

| Reagent/Resource | Reference or Source | Identifier or Catalog Number |
|---|---|---|
| **Experimental models** | | |
| HEK293T (PAC-KO) | Chen et al, 2025 | https://doi.org/10.1038/s41594-025-01495-8 |
| HEK293S GnTI⁻ suspension cells | Chen et al, 2025 | https://doi.org/10.1038/s41594-025-01495-8 |
| Sf9 cells | Chen et al, 2025 | https://doi.org/10.1038/s41594-025-01495-8 |
| *Scolopendra subspinipes mutilans* | Chuzhou Municipal Dafeng Breeding Co., Ltd., China | N/A |

| Reagent/Resource | Reference or Source | Identifier or Catalog Number |
| --- | --- | --- |
| **Recombinant DNA** | | |
| pcDNA3.1-PDPNaC1 | This study | N/A |
| pEG-BM-PDPNaC1 | This study | N/A |
| pANAP | Addgene | N/A |
| **Antibodies** | | |
| Mouse anti-PDPNaC1 | Biodragon | N/A |
| Rabbit anti-PIEZO2 | Thermo Fisher Scientific | Cat #PA5-111032 |
| Rabbit anti-GAPDH | Proteintech Group, Inc. | Cat #81640-5-RR |
| Goat anti-mouse | Proteintech Group, Inc. | Cat #RGAM001 |
| Goat anti-rabbit | Proteintech Group, Inc. | Cat #RGAR001 |
| **Oligonucleotides and other sequence-based reagents** | | |
| PCR primers | This study | Appendix Table S3 |
| **Chemicals, Enzymes and other reagents** | | |
| Protease Inhibitor Cocktail | MedChemExpress | HY-K0010-1 |
| Phosphatase Inhibitor Cocktail I | MedChemExpress | HY-K0021-1 |
| NcmECL Ultra | NCM Biotech | P10100 |
| Alexa Fluor® 488 Conjugate anti-mouse IgG | Cell Signaling Technology | Cat #4408S |
| Cy™3 anti-rabbit IgG | Jackson ImmunoResearch Inc. | Cat #111-165-003 |
| DAPI | Servicebio | 2503F001 |
| O.C.T | SAKURA | Cat #4583 |
| 4% Paraformaldehyde Fixative | Biosharp | Cat #BL539A |
| Methyl red indicator | Yuanye | Cat #R22089 |
| DMEM basic | Thermo Fisher Scientific | Cat #C11995500BT |
| 0.25% Trypsin-EDTA | Thermo Fisher Scientific | Cat #25200-072 |
| Fetal Bovine Serum | Thermo Fisher Scientific | Cat #A5669701 |
| Lipofectamine 3000 | Thermo Fisher Scientific | Cat #L3000015 |
| Zinc chloride | Macklin | Cat #7646-85-7 |
| DDM (n-dodecyl-β-D-maltopyranoside) | Anatrace | Cat #M2503 |
| CHS (cholesteryl hemisuccinate) | Anatrace | Cat #CH210 |
| LMNG (lauryl maltose neopentyl glycol) | Anatrace | Cat #NG310 |
| Sodium butyrate | Sangon | Cat #A510838 |
| Strep-Tactin Sepharose beads | Smart-Lifesciences | Cat #SA053100 |
| L-Anap | AsisChem, Inc. | N/A |
| Sodium chloride | Sigma | Cat #S6191 |
| Potassium chloride | Sigma | Cat #P9541 |
| Cesium chloride | Sigma | Cat #V900481 |
| Magnesium chloride hexahydrate | Sigma | Cat #V900020 |
| MES monohydrate | Sigma | Cat #V900336 |
| HEPES | Sigma | Cat #V900477 |
| D-(+)-Glucose | Sigma | Cat #G8270 |
| N-Methyl-D-glucamine | Sigma | Cat #M2004 |
| Sodium hydroxide | Sigma | Cat #S5881 |
| Potassium hydroxide | Sigma | Cat #P5958 |
| Cesium hydroxide solution | Sigma | Cat #232068 |
| **Software** | | |
| Cell Ranger (v 7.0.1) | 10x Genomics | https://support.10xgenomics.com/single-cell-gene-expression/software/downloads/latest |
| R (v 4.3.1) | The R Project | https://www.r-project.org |
| Seurat (v 5.0.1) | Satija Lab | https://satijalab.org/seurat |
| cryoSPARC v4 | Structura Biotechnology | https://cryosparc.com/ |
| Coot | MRC | N/A |
| Titan Krios | FEI | N/A |
| PyMOL v2.5.2 | Schrödinger, LLC | https://pymol.org |
| ChimeraX v1.8 | UCSF | https://www.cgl.ucsf.edu/chimerax |
| **Other** | | |
| Illumina NovaSeq X Plus | Illumina | |

## Animals

All experiments involving animals conformed to the recommendations in the Guide for the Care and Use of Laboratory Animals of Northeast Forestry University. All experimental procedures were approved by the Animal Care and Use Committee at Northeast Forestry University (approval ID 2023072). All efforts were made to minimize sample size and animal suffering. Adult centipedes (*Scolopendra subspinipes mutilans*) were purchased from Chuzhou Municipal Dafeng Breeding Co., Ltd., China.

## Single-nucleus RNA sequencing and analysis

Antennae tissues were dissected from anesthetized centipedes, flash-frozen in liquid nitrogen, and shipped on dry ice to Novogene

Corporation (Tianjin, China) for nuclei isolation. Frozen tissues were homogenized in 5 mL lysis buffer using 10 loose and 5 tight pestle strokes, then filtered through 30-μm strainers. Nuclei were purified via OptiPrep density gradient centrifugation (29% cushion, 10,000 rpm, 30 min, 4 °C), quantified using LUNA counter with AO/PI staining, and resuspended in nuclei buffer (~$1 \times 10^6$ nuclei/mL). Quality control included microscopic integrity assessment and viability >85%. Single-nucleus libraries were constructed using the Chromium Single Cell 3' Reagent Kit v3.1 (10x Genomics) on a 10x Chromium Controller (10x Genomics). Poly(A) RNA capture and cDNA synthesis following the 10x Genomics 3' gene expression workflow. Sequencing was performed on an Illumina NovaSeq X Plus platform (150 bp paired-end). Nuclei isolation from antennal tissues was conducted by Novogene Corporation (Tianjin, China) under standard service agreements to ensure data quality and consistency. Specimen collection followed institutional guidelines for invertebrate research. Raw sequencing data are deposited in GEO (GSE276298).

## Data analysis and cell annotation

Raw reads were aligned to the *S. s. mutilans* genome using Cell Ranger (v 7.0.1). Data analysis was performed in R (v 4.3.1) using Seurat (v 5.0.1) (Butler et al, 2018; Stuart et al, 2019). Low-quality nuclei (<200 detected genes or >10% mitochondrial reads) were excluded. Doublets were removed with DoubletFinder v 2.0.4 (McGinnis et al, 2019) (simulated pseudo-doublets, 5% exclusion threshold). Highly variable genes (top 2000) were selected via variance-stabilizing transformation. Principal component analysis (PCA) was applied to the top 10 dimensions, followed by UMAP embedding and graph-based clustering (resolution = 0.3). Cluster-specific marker genes were identified with the Wilcoxon rank-sum test (min.pct = 0.01, $\log_2$ fold-change > 0.1). Cell types were annotated by cross-referencing markers against published scRNA-seq datasets (Li et al, 2022a; Li et al, 2022b) and the CellMarker 2.0 database (Hu et al, 2023).

## Polyclonal antibody preparation against PDPNaC1

A custom polyclonal antibody targeting PDPNaC1 (residues 193–208) was produced by Biodragon (Suzhou, China). Female Balb/C mice ($n = 7$, 6–8 weeks old) were used for immunization with KLH-conjugated peptide (10 mg total dose). The animals used in this study were compliant with the ethical standards, and the usage of the animals was approved under the animal use license No. SYXK (Su) 2021-0031. Serum was collected 24 days after immunization for further purification following the company's standard protocols. The purified antibody exhibited a titer >128,000 and a concentration of 1.28 mg/mL. Antibodies were analyzed by Western blot and Immunofluorescence (Appendix Fig. S13).

## Antibody validation by western blot

HEK293T cells were transfected with a PDPNaC1 expression plasmid in 10 cm dishes. After 24 h, cells were detached with trypsin, washed with PBS, and pelleted at 600 rpm for 5 min. Cells were lysed in 100 μl RIPA (89900, Thermo Fisher Scientific, USA) buffer supplemented with 1% Protease Inhibitor Cocktail (HY-K0010-1, MedChemExpress, USA) and 1% Phosphatase Inhibitor Cocktail I (HY-K0021-1, MedChemExpress, USA). Lysates were

incubated on ice for 40 min and cleared by centrifugation at 12,000 rpm for 10 min. Untransfected HEK293T cells were used as negative control. Samples were mixed with 5× SDS loading buffer, boiled at 95 °C for 5 min, and resolved by SDS-PAGE. Protein samples (20 μL per lane) were separated on 10% polyacrylamide gels prepared using the One-Step PAGE Gel Fast Preparation Kit (E303-01, Vagyme, China) with 1-mm, 10-well combs. Electrophoresis was performed using a Tanon EPS 300 system at 80 V (stacking gel) and 120 V (resolving gel) until the dye front reached the bottom of the gel. A pre-stained protein marker (MP201, Vagyme, China) was used for molecular weight estimation. Proteins were transferred to methanol-activated polyvinylidene difluoride (PVDF) membranes via wet transfer at a constant current of 200 mA for ~70 min using the same electrophoresis system. Membranes were blocked in 5% bovine serum albumin (BSA) in TBST for 1–2 h at room temperature and incubated with anti-PDPNaC1 (mouse, 1:500, Biodragon, China) for 1 h. After three washes in TBST, membranes were incubated for 1–2 h at room temperature with HRP-conjugated anti-mouse secondary antibody (1:10,000, RGAM001, Proteintech Group, Inc., USA) for 1 h. After enhanced chemiluminescence (NcmECL Ultra, P10100, NCM Biotech, China) detection, membranes were rinsed twice in TBST (10 min each) and directly incubated (without stripping) with anti-GAPDH (rabbit, 1:10,000, 81640-5-RR, Proteintech Group, Inc., USA) for 1 h. Membranes were then washed 3–5 times in TBST (10 min each), followed by anti-rabbit HRP secondary antibody (1:10,000, RGAR001, Proteintech Group, Inc., USA) for 1 h. Blots were developed after each antibody incubation step. All antibodies were incubated at room temperature with gentle agitation (80 rpm). Images were acquired with a chemiluminescence imaging system (Tanon 5200 Multi, China). No membrane trimming was performed for imaging.

## Antibody validation by immunofluorescence in cultured cells

HEK293T cells were transfected with plasmids encoding PDPNaC1 using Lipofectamine 3000 (Thermo Fisher Scientific) for 24 h in 3.5 cm culture dishes. Transfected cells were then harvested by trypsinization and seeded onto 1.0 cm glass coverslips. Untransfected HEK293T cells were used as negative control. After attachment, cells were fixed in 4% paraformaldehyde for 15 min at room temperature and permeabilized with 0.1% Triton X-100 for 10 min. The samples were blocked with 5% bovine serum albumin (BSA) in PBS for 1 h and then incubated with a mouse primary anti-PDPNaC1 (1:200, Biodragon, China) overnight at 4 °C. After washing with PBS (3 × 5 min), cells were incubated with an Alexa Fluor® 488 Conjugate anti-mouse IgG (1:500, 4408S, Cell Signaling Technology, USA) for 1 h at room temperature in the dark. Nuclei were counterstained with DAPI (1 μg/mL, 2503F001, Servicebio, China) for 10 min. After final washes, the coverslips were inverted onto 3.5 cm circular glass-bottom dishes (BS-20-GJM, Biosharp, China) for imaging. Images were acquired using an Olympus IX83 confocal microscope without post-acquisition processing.

## Histological analysis

Centipede antennae were washed with PBS, embedded in optimal cutting temperature (OCT) compound, and flash-frozen in liquid

nitrogen. Samples were sectioned into 8μm slices using a microtome (Leica RM2235). Sections were fixed in 4% paraformaldehyde (PFA) for 15 min at room temperature and washed three times with PBS (5 min each). Permeabilization was performed with 0.5% Triton X-100 for 10 min, followed by blocking in 5% bovine serum albumin (BSA) in 0.3 M glycine PBS at room temperature for 1 h. Primary antibodies-anti-PIEZO2 (1:200, PA5-111032, Thermo Fisher Scientific, USA) and anti-PDPNaC1 (1:200, Biodragon, China) were diluted in blocking buffer and incubated with sections overnight at 4 °C. After three washes with PBST (PBS with 0.1% Tween-20), sections were incubated with secondary antibodies for 1 h at room temperature. The secondary antibodies used were Cy™3 anti-rabbit IgG (1:500, 111-165-003, Jackson ImmunoResearch Inc., USA) and Alexa Fluor® 488 Conjugate anti-mouse IgG (1:500, 4408S, Cell Signaling Technology, USA). After counterstaining with DAPI, fluorescence images were acquired using a confocal laser scanning microscope (Olympus IX83).

## Immunofluorescence colocalization analysis

Quantitative colocalization analysis was performed using FIJI (ImageJ, NIH) with the Coloc2 plugin. Immunostained images were first preprocessed by background subtraction. Three representative regions of interest (ROIs) were manually selected from each image to capture areas with clear and overlapping signals of the two fluorophores. The selected ROIs were then combined into a single dataset for statistical analysis. Colocalization was assessed using the Coloc2 plugin, with calculation of Pearson's correlation coefficient (PCC) and Manders' overlap coefficients (M1 and M2) to evaluate the degree of spatial correlation between the two fluorescent signals. The analysis was performed using default settings, including Costes' automatic thresholding to reduce background influence. The final reported colocalization metrics represent the average values computed from the combined ROIs across multiple images. Fluorescence colocalization data points were exported as CSV files from FIJI. To remove background noise, points with zero counts (count = 0) were excluded from the analysis. The Y-axis coordinates were inverted to match the image coordinate system by subtracting the original Y values from the maximum Y coordinate.

## Scanning electron microscopy of centipede antennae

Centipedes were euthanized by decapitation, and antennae were immediately excised and snap-frozen in liquid nitrogen. Samples were freeze-dried at −55 °C for 48 h using a freeze dryer (Alpha 1–4 LSCbasic, CHRIST, Germany). Before imaging, specimens were sputter-coated with a thin layer of platinum. Scanning electron microscopy was performed using a scanning electron microscope (Apreo S HiVac, Thermo Scientific, USA) under the following conditions: standard mode, 5.00 kV accelerating voltage, 0.10 nA beam current, 1000× magnification, 11.4 mm working distance, ETD detector, and 207 μm horizontal field width. Images were acquired using Microscope Control (v 13.5.0) software.

## Animal behavior and pH droplet assay

Centipedes were individually placed in transparent Plexiglas chambers (30.5 × 8.5 × 30.5 cm) at 25 °C. Each individual was allowed to acclimate for 2 h before recording. For gas stimulation, a 500 mL plastic bottle with perforated sides for ventilation was used. A pipette tip was inserted into the center of the cap to deliver either ambient air or 95% $CO_2$. Two stimulation protocols were used: CA/CC (Continuous Air/$CO_2$): Continuous delivery of air or $CO_2$ for 10 min while recording antennal swing frequency. IA/IC (Intermittent Air/$CO_2$): Alternating 30-s pulses of air or $CO_2$ every 30 s for 10 min, with continuous behavioral recording throughout. All recordings were captured using a Canon EOS 500D digital camera. Antennal swing frequencies were analyzed using the Kruskal–Wallis rank-sum test, followed by Dunn's post hoc test for multiple comparisons (R package "dunn.test"). Statistical significance was accepted at $p < 0.05$.

For the droplet-based pH assay, a transparent Petri dish containing a methyl red indicator solution was placed under a stereomicroscope (Leica M205 FCA). Air or 95% $CO_2$ was directed over the droplet to observe and photograph pH-dependent color changes. The relationship between droplet diameter and pH transition time was analyzed in R (v 4.3.1). After applying natural logarithmic (ln) transformation to both variables, a linear fit was performed according to the model:

$$\ln(Y) = \ln(A) + b \times \ln(X)$$

The fitted parameters $A$ and $b$ defined the power function:

$$Y = A \times X^b$$

where $Y$ is the time for pH transition, $A$ is the pre-factor, $b$ is the exponent, $X$ is the drop diameter, and this function was used to predict pH transition times for smaller droplets.

## Molecular biology, cell preparation, and transfection

The full-length, codon-optimized cDNA of PDPNaC1 was synthesized by Sangon Biotech (Shanghai, China) and subcloned into the pcDNA3.1 vector. A plasmid encoding chicken ASIC1a (cASIC1a, Gene ID: 426883) was also used in this study. Point mutations were introduced using the Fast Mutagenesis Kit V2 (Vazyme, China) according to the manufacturer's instructions. The mutations were sequenced to confirm the accuracy of the constructs. Primer sequences used for mutagenesis are listed in Appendix Table S3. PAC knockout HEK293T cells (Chen et al, 2025) were cultured in Dulbecco's modified Eagle medium (DMEM) supplemented with 10% fetal bovine serum and 1% penicillin/streptomycin at 37 °C in 5% $CO_2$. Cells were transfected with the channel constructs together with enhanced green fluorescent protein (eGFP) plasmid using Lipofectamine 3000 (Thermo Fisher Scientific, USA), according to the manufacturer's instructions. Twenty-four hours after transfection, fluorescent cells were selected for patch-clamp recordings. To assess whether endogenous ASIC channels affect the recording of PDPNaC1 currents, we detected ASIC-like currents. Although a fraction of cells expressed endogenous ASIC channels, their presence did not affect the recording of PDPNaC1 currents, owing to the rapid desensitization of ASIC channels at low pH (Appendix Fig. S14).

## Electrophysiology

Whole-cell patch-clamp recordings were performed using an EPC10 amplifier controlled by PatchMaster software (HEKA Elektronik,

Lambrecht, Germany). Patch pipettes were pulled from borosilicate glass and fire-polished to a resistance of ~6 MΩ. The internal (pipette) solution contained 140 mM KCl, 2 mM MgCl$_2$, 5 mM EGTA, 10 mM HEPES, and 10 mM glucose (pH 7.4, adjusted with KOH). The standard external (bath) solution contained 140 mM NaCl, 10 mM HEPES, and 10 mM glucose (pH 7.4, adjusted with NaOH). Since PDPNaC1 is impermeable to divalent cations such as Ca$^{2+}$ and Mg$^{2+}$, and Ca$^{2+}$ could modulate the channel's proton sensitivity, we opted to exclude them from the standard extracellular solution to simplify the ionic environment and facilitate interpretation of monovalent cation selectivity. Bath solutions at different pH levels were adjusted using HCl. The membrane potential was held at −80 mV throughout the experiments.

For permeable ion composition experiments of PDPNaC1, both internal and external solutions contained 140 mM NMDG-Cl and 10 mM HEPES (pH 7.4, adjusted with HCl). After establishing the whole-cell configuration, the bath solution was replaced with 140 mM NMDG-Cl and 10 mM MES (pH 3.0, adjusted with HCl). To assess ion permeability upon proton binding, the bath solution was subsequently exchanged for 140 mM NaCl, KCl, CsCl, or 110 mM MgCl$_2$ in 10 mM MES (pH 3.0, adjusted with HCl). To assess ion permeability upon proton dissociation, a stable current was first elicited using pH 3.0 NMDG-Cl. The proton removal-induced current was then recorded by replacing the solution with 140 mM NaCl, KCl, CsCl, or 110 mM MgCl$_2$ in 10 mM MES (pH 7.4, adjusted with NaOH, KOH, CsOH, or Mg(OH)$_2$, respectively). The holding potential was maintained at −80 mV.

For cation permeability experiments of PDPNaC1 and its mutants, both pipette and bath solutions initially contained 140 mM NaCl and 10 mM MES (pH 7.4, adjusted with NaOH). After whole-cell access was achieved, the bath solution was replaced with 140 mM NaCl, KCl, or CsCl in 10 mM MES (pH 3.0, adjusted with HCl), followed by substitution with the same salts in 10 mM MES (pH 7.4, adjusted with NaOH, KOH, or CsOH). Currents were recorded using voltage ramps from −60 mV to +60 mV over a duration of 120 ms. The current induced by proton removal was shown and used for ion selectivity analysis.

For cation permeability experiments of cASIC1a, the pipette solution contained 140 mM KCl, 5 mM EGTA, 10 mM MES (pH 7.4, adjusted with KOH). The bathing solution contained 140 mM NaCl, 10 mM MES (pH 7.4, adjusted with NaOH). After the whole-cell configuration was obtained in standard solution, the bathing solution was changed to 140 mM NaCl (or KCl or CsCl), 10 mM MES (pH 6.0, adjusted with HCl). The reversal potential was determined using voltage ramps from −80 mV to +80 mV over a duration of 160 ms.

For the voltage dependence of PDPNaC1 and mutants, the pipette solution contained 140 mM NaCl, 5 mM EGTA, 10 mM HEPES (pH 7.4, adjusted with NaOH). The bathing solution contained 140 mM NaCl, and 10 mM HEPES (pH 7.4, adjusted with NaOH). After whole-cell access was achieved, the bath solution was replaced with 140 mM NaCl, 10 mM MES (pH 3.0, adjusted with HCl), followed by substitution with bathing solution. The membrane potential was held at −30 mV, −55 mV or −80 mV throughout the experiments.

## Protein expression and purification

The *PDPNaC1* gene was cloned into a pEG-BM vector with a C-terminus Flag tag and two tandem Strep tags. The construct was heterologously expressed in HEK293S GnTI$^-$ suspension cells (Life Technologies) using the BacMam system (Thermo Fisher Scientific). P3 baculovirus, generated from Sf9 cells using standard protocol, was used to transfect HEK293S GnTI$^-$ cells at a density of $4.0 \times 10^6$ cells/mL. After 12 h, 10 mM sodium butyrate was added, and the cells were cultured at 30 °C for an additional 48 h to enhance expression. Cells were harvested by centrifugation at 3000 rpm for 10 min at 4 °C.

Cells were resuspended in lysis buffer A (10 mM MES, pH 7.4, 140 mM NaCl) supplemented with a protease inhibitor (2 μg/mL DNase I, 0.5 μg/mL pepstatin, 1 μg/mL aprotinin, and 1 mM phenylmethylsulfonyl fluoride), and lysed by sonication on ice. The PDPNaC1 protein was extracted with 1.0% (w/v) n-dodecyl-β-D-maltopyranoside (DDM, Anatrace) and 0.2% (w/v) cholesteryl hemisuccinate (CHS, Anatrace) at 100 rpm for 2 h at 4 °C. After centrifugation at 46,000 rpm for 45 min at 4 °C, the supernatant was incubated with Strep-Tactin beads for 1 h at 4 °C. The beads were washed with buffer A containing 0.05% (w/v) lauryl maltose neopentyl glycol (LMNG, Anatrace) and 0.001% CHS. Protein was eluted with 5 mL of buffer A (pH 8.0, adjusted with NaOH) containing 0.005% LMNG, 0.001% CHS, and 10 mM desthiobiotin. The eluate was concentrated to 1 mL, and centrifuged at 14,000 rpm for 10 min. The supernatant was subjected to size-exclusion chromatography (SEC) using a Superose 6 10/300 GL column, equilibrated with buffer A containing 0.005% LMNG and 0.001% CHS. The peak fraction was concentrated to 4.0 mg/mL for cryo-EM analysis.

## Cryo-EM data acquisition

Cryo-EM data were collected at the Center of Cryo-Electron Microscopy, Zhejiang University. Three microliters of PDPNaC1 samples were applied to glow-discharged holey carbon grids (Quantifoil Au R1.2/1.3, 300-mesh), blotted for 5 s under 100% humidity at 4 °C, and vitrified in liquid ethane using a Vitrobot (FEI). Grids were loaded into a Titan Krios transmission electron microscope (FEI) operating at 300 kV and equipped with a K2 Summit direct electron detector (Gatan). Automated data collection was performed using SerialEM. Images were acquired at a calibrated magnification of 49,310×, corresponding to a pixel size of 1.014 Å. Each micrograph was dose-fractionated into 40 frames with a total exposure of 8 s at a dose rate of 8 e$^-$/pixel/s, yielding a total dose of ~55.5 e$^-$/Å$^2$.

## Data processing

Movie stacks were motion-corrected, and contrast transfer function (CTF) estimation was performed in cryoSPARC v4. A total of 891,249 particles were auto-picked using the Blob picker from 1938 micrographs. After 2D classification, selected particles were subjected to ab initio reconstruction with C1 symmetry. One class showed clear secondary structural features and was refined using non-uniform refinement with C3 symmetry. The final 3D reconstruction, based on 121,831 particles, yielded a map at 3.03 Å resolution.

## Model building, refinement, and validation

The atomic model of PDPNaC1 was built de novo in Coot using the 3.03 Å density map. Side-chain assignment was guided by the

densities of bulky residues (Phe, Trp, Tyr, and Arg). The model was refined against the map using real-space refinement in PHENIX with secondary structure restraints and non-crystallographic symmetry. Residues 35–399 were confidently modeled. Validation was performed using MolProbity. Structural figures were prepared using PyMOL, ChimeraX, or Coot. Additional details are provided in Appendix Table S4.

## Site-directed ANAP fluorescence recordings

L-ANAP methyl ester was purchased from AsisChem, and the pANAP vector was obtained from Addgene. ANAP was incorporated into the PDPNaC1 channel by introducing a TAG amber-stop codon mutation. During transfection, PAC knockout HEK293T cells were co-transfected with 1 µg pANAP vector and 3 µg ANAP mutant plasmid of PDPNaC1. A final concentration of 20 µM ANAP was directly added to the culture medium. This method enabled efficient delivery of ANAP into cells and its site-specific incorporation into the target channel proteins, resulting in robust and position-specific fluorescence signals (Lu et al, 2022; Yang et al, 2020). Cells expressing ANAP-incorporated PDPNaC1 channels were digested 24 h post-transfection, resuspended in fresh medium, and allowed to adhere to the glass slides for electrophysiological recording or imaging.

ANAP fluorescence images of cells were captured using an Olympus IX71 microscope. The emission spectrum of ANAP was obtained using an Acton ARC-SP 2156 Imaging Spectrograph (Princeton) in conjunction with an Orca R2 C10600-10B CCD camera (Hamamatsu). ANAP was excited using the X-Cite XYLIS light source with a 340–390 nm excitation filter and a 420LP emission filter (>420 nm). The ANAP emission peak was measured by fitting the recorded emission spectrum with a skewed Gaussian distribution in Igor Pro version 6.37.

## Fluorescence emission spectrum measurement of ANAP under different solvents and pH conditions

To assess the fluorescence emission properties of the unnatural amino acid ANAP under various environmental conditions, two solvent systems were prepared: 50% DMSO:50% PBS and 80% DMSO:20% PBS (v/v). The pH of each solution was adjusted to 7.0, 5.0, or 3.0 using HCl. ANAP was dissolved in each solution at a final concentration of 2 µM. Fluorescence emission spectra were recorded using an RF-6000 spectrofluorophotometer (Shimadzu, S/N: A40245601180SA) operated with LabSolutions RF software (version 1.15). The excitation wavelength was set to 345 nm, and emission spectra were collected from 365 nm to 670 nm with a data interval of 1.0 nm. Both excitation and emission slit widths were set to 5.0 nm. The scanning speed was 2000 nm/min. All measurements were performed at room temperature in quartz cuvettes.

## Electrostatic surface potential calculation

The atomic structure of PDPNaC1 at pH 8.0 was used to calculate electrostatic surface potentials. Hydrogen atoms were added, and protonation states were assigned using the PDB2PQR server (https://server.poissonboltzmann.org/pdb2pqr), with the AMBER force field and pH set to 8.0. The resulting PQR file was then used as input for the APBS solver to compute electrostatic potentials using default settings. The electrostatic map (.dx file) was visualized

in ChimeraX 1.8 by mapping the potential onto the molecular surface using the built-in Coulombic Surface Coloring tool. The color scale was set to ±20 kT/e, with red indicating negative and blue indicating positive potential.

## Statistical analysis

Phylogenetic analysis of protein sequences was carried out using MEGA X (Kumar et al, 2018). Electrophysiological data were analyzed with Igor Pro (WaveMetrics, version 6.37) and Prism (GraphPad, version 8.0.1). Behavioral data from animal assays were processed using R (v 4.3.1). The antenna swing frequency of centipedes to different chemical microenvironments was compared by the Kruskal–Wallis rank-sum test. The R package "dunn.test" for the Kruskal–Wallis rank-sum test was employed. Statistical significance was accepted at a level of $p < 0.05$.

Half-maximal activating concentration ($EC_{50}$) for agonist-induced responses was determined by fitting the dose–response curves to the Hill equation:

$$\frac{I_A}{I_{max}} = \frac{[A]^n}{EC_{50}^n + [A]^n}$$

where $I_A$ is the current amplitude at agonist concentration $[A]$, $I_{max}$ is the maximal current response, $EC_{50}$ is the agonist concentration producing half-maximal activation, and $n$ is the Hill coefficient.

Half-maximal inhibitory concentration ($IC_{50}$) was determined by fitting the concentration-response curves to the Hill equation:

$$\frac{I_x}{I_{max}} = 1 - \frac{[X]^n}{IC_{50}^n + [X]^n}$$

where $I_X$ is the remaining current at inhibitor concentration $[X]$, $I_{max}$ is the current without inhibitor, $IC_{50}$ is the inhibitor concentration producing half-maximal inhibition, and $n$ is the Hill coefficient.

Permeability ratios for monovalent to cation $Na^+$ ($P_X/P_{Na^+}$) were calculated using the Goldman-Hodgkin-Katz equation:

$$P_{(X)}/P_{(Na^+)} = \exp\left(\frac{\Delta VrevF}{RT}\right)$$

where $Vrev$ presents the reversal potential, $F$ represents Faraday's constant, $R$ is the universal gas constant, and $T$ is absolute temperature.

All values are given as mean ± SEM, with the number of measurements indicated ($n$). Statistical significance was determined using the t test (two-tailed, unpaired). Significance is denoted as follows: $*p < 0.05$, $**p < 0.01$, $***p < 0.001$ and $****p < 0.0001$, n.s., no significant.

# Data availability

The raw data generated in this study have been deposited in the GEO database under accession code GSE276298 and in the SRA database under accession code PRJNA1156367. PDPNaC1 cDNA sequence data have been deposited in Genbank (ID: PQ282429). Structure coordinates and cryo-EM density maps have been deposited in the Protein Data Bank under accession numbers

9JF7 and EMD-61429 for PDPNaC1. Accessible via the following link: URL: https://www.ncbi.nlm.nih.gov/geo/query/acc.cgi?acc=GSE276298. URL: https://www.ncbi.nlm.nih.gov/bioproject/?term=PRJNA1156367. URL: https://www.ncbi.nlm.nih.gov/nuccore/PQ282429. PDB: https://doi.org/10.2210/pdb9JF7/pdb. EMDB: https://www.ebi.ac.uk/emdb/EMD-61429. The plasmids used in this study can be provided by YW. Requests for the plasmids should be submitted to the College of Wildlife and Protected Area, Northeast Forestry University, China.

The source data of this paper are collected in the following database record: biostudies:S-SCDT-10_1038-S44319-025-00606-2.

## Peer review information

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

## Acknowledgements

This work was supported by grants from the National Key Research and Development Program of China (2023YFF1304900), the National Natural Science Foundation of China (32430012 and 32170486), the National Key Research and Development Program of Heilongjiang province (2024ZXDXC28), and the Supported by the Fundamental Research Funds for the Central Universities (2572022DS03) to SY; from the National Natural Science Foundation of China (32370437), the Fundamental Research Funds for the Central Universities (2572023CT13) and the supported by the Longjiang Science and Technology Talents Spring Goose Support Program (CYQN24058) to YW; from the National Natural Science Foundation of China (32370505) and Heilongjiang Province (YQ2022H001), the Fundamental Research Funds for the Central Universities (2572025JT07), and the National Key Research and Development Program of China (2023YFF1305000) to XL; from the Fundamental Research Funds for the Central Universities (2572025AW13) to LY.

## Author contributions

**Wenqi Dong**: Data curation; Formal analysis; Validation; Writing—original draft. **Licheng Yuan**: Software; Formal analysis; Validation. **Jiangming Shang**: Software; Formal analysis; Visualization. **Fan Yang**: Conceptualization; Data curation; Methodology. **Shilong Yang**: Conceptualization; Supervision; Writing—original draft; Writing—review and editing. **Xiancui Lu**: Conceptualization; Data curation; Supervision; Writing—original draft; Writing—review and editing. **Qian Wang**: Formal analysis; Validation. **Anna Luo**: Formal analysis; Visualization. **Jiheng Geng**: Formal analysis; Visualization. **Jiatong Cheng**: Software; Formal analysis; Visualization. **Runze Li**: Software; Formal analysis. **Yunfei Wang**: Conceptualization; Data curation; Funding acquisition; Writing—original draft; Writing—review and editing.

Source data underlying figure panels in this paper may have individual authorship assigned. Where available, figure panel/source data authorship is listed in the following database record: biostudies:S-SCDT-10_1038-S44319-025-00606-2.

## Disclosure and competing interests statement

The authors declare no competing interests.

