## [Peer Review File · EMBO Reports]

A Proton-Gated Channel Identified in the Centipede Antenna

Wenqi Dong, Licheng Yuan, Jiangming Shang, Fan Yang, Shilong Yang, Xiancui Lu, Qian Wang, Anna Luo, Jiheng Geng, Jiatong Cheng, Runze Li, and Yunfei Wang

Corresponding author(s): Yunfei Wang (wangyunfei@nefu.edu.cn)

Review Timeline:

Submission Date:	27th May 25
Editorial Decision:	8th Jul 25
Revision Received:	4th Aug 25
Editorial Decision:	12th Sep 25
Revision Received:	23rd Sep 25
Accepted:	7th Oct 25

Editor: *Martina Rembold*

Transaction Report:

Dear Dr. Wang

Thank you for the submission of your research manuscript to our journal. We have now received the full set of referee reports that is copied below.

As you will see, the referees acknowledge that the findings are interesting and that the conclusions are overall supported by the data presented but they also raise a number of concerns and have suggestions how to further strengthen the data.

Given the constructive comments, we would like to invite you to revise your manuscript with the understanding that the referee concerns must be fully addressed and their suggestions taken on board. Please address all referee concerns in a complete point-by-point response. Acceptance of the manuscript will depend on a positive outcome of a second round of review. It is EMBO Reports policy to allow a single round of revision only and acceptance or rejection of the manuscript will therefore depend on the completeness of your responses included in the next, final version of the manuscript.

We realize that it is difficult to revise to a specific deadline. In the interest of protecting the conceptual advance provided by the work, we recommend a revision within 3 months (October 7th). Please discuss the revision progress ahead of this time with the editor if you require more time to complete the revisions.

I am also happy to discuss the revision further via e-mail or a video call, if you wish.

=====
IMPORTANT NOTE:

We perform an initial quality control of all revised manuscripts before re-review. Your manuscript will FAIL this control and the handling will be delayed IN CASE the following APPLIES:

- 1) A data availability section providing access to data deposited in public databases is missing. If you have not deposited any data, please add a sentence to the data availability section that explains that.
- 2) Your manuscript contains statistics and error bars based on $n=2$. Please use scatter blots in these cases. No statistics should be calculated if $n=2$.

=====
When submitting your revised manuscript, we will require:

- 1) a .docx formatted version of the manuscript text (including legends for main figures, EV figures and tables). Please make sure that the changes are highlighted to be clearly visible.
- 2) individual production quality figure files as .eps, .tif, .jpg (one file per figure). Please download our Figure Preparation Guidelines (figure preparation pdf) from our Author Guidelines pages <https://www.embopress.org/page/journal/14693178/authorguide> for more info on how to prepare your figures.
- 3) a .docx formatted letter INCLUDING the reviewers' reports and your detailed point-by-point responses to their comments. As part of the EMBO Press transparent editorial process, the point-by-point response is part of the Review Process File (RPF), which will be published alongside your paper.
- 4) a complete author checklist, which you can download from our author guidelines (<<https://www.embopress.org/page/journal/14693178/authorguide>>). Please insert information in the checklist that is also reflected in the manuscript. The completed author checklist will also be part of the RPF.
- 5) Please note that all corresponding authors are required to supply an ORCID ID for their name upon submission of a revised

manuscript (<<https://orcid.org/>>). Please find instructions on how to link your ORCID ID to your account in our manuscript tracking system in our Author guidelines (<<https://www.embopress.org/page/journal/14693178/authorguide#authorshipguidelines>>)

6) We replaced Supplementary Information with Expanded View (EV) Figures and Tables that are collapsible/expandable online. A maximum of 5 EV Figures can be typeset. EV Figures should be cited as 'Figure EV1, Figure EV2' etc... in the text and their respective legends should be included in the main text after the legends of regular figures.

7) Data deposited in public repositories: The accession numbers and database should be listed in a formal "Data Availability " section (placed after Materials & Method) that follows the model below (see also <<https://www.embopress.org/page/journal/14693178/authorguide#dataavailability>>). Please note that the Data Availability Section is restricted to new primary data that are part of this study.

Data availability

Additional information on source data and instruction on how to label the files are available <<https://www.embopress.org/page/journal/14693178/authorguide#sourcedata>>

10) Figure legends and data quantification:

- the name of the statistical test used to generate error bars and P values,
 - the EXACT p-values,
 - the number (n) of independent experiments (please specify technical or biological replicates) underlying each data point,
 - the nature of the bars and error bars (s.d., s.e.m.)
- If the data are obtained from n {less than or equal to} 5, show the individual data points in addition to the SD or SEM.
- If the data are obtained from n {less than or equal to} 2, use scatter blots showing the individual data points.

11) Our journal encourages inclusion of *data citations in the reference list* to directly cite datasets that were re-used and obtained from public databases. Data citations in the article text are distinct from normal bibliographical citations and should

directly link to the database records from which the data can be accessed. In the main text, data citations are formatted as follows: "Data ref: Smith et al, 2001" or "Data ref: NCBI Sequence Read Archive PRJNA342805, 2017". In the Reference list, data citations must be labeled with "[DATASET]". A data reference must provide the database name, accession number/identifiers and a resolvable link to the landing page from which the data can be accessed at the end of the reference. Further instructions are available at <<https://www.embopress.org/page/journal/14693178/authorguide#referencesformat>>.

12) All Materials and Methods need to be described in the main text using our 'Structured Methods' format. According to this format, the Methods section includes a Reagents and Tools Table (listing key reagents, experimental models, software and relevant equipment and including their sources and relevant identifiers) followed by a Methods and Protocols section describing the methods, ideally using a step-by-step protocol format. The aim is to facilitate adoption of the methodologies across labs. Please download and fill our Reagents and Tools Table template (.docx), which you can find in our author guidelines:

13) As part of the EMBO publication's Transparent Editorial Process, EMBO Reports publishes online a Review Process File to accompany accepted manuscripts. This File will be published in conjunction with your paper and will include the referee reports, your point-by-point response and all pertinent correspondence relating to the manuscript.

Yours sincerely,

=====

Referee #1:

In this study, Dong and colleagues cloned an apparent ASIC homolog in centipede, solved its structure with cryo-EM, characterized its function and delineated the structure-function correlates, and proposed that the channel serves as a CO₂ sensor. Through this effort, they identified the channel's activation gate, the ion selectivity filter, and a proton binding site that gives the channel its prominent proton-dependent conductance. This is a large body of work that presents an interesting new ion channel with some peculiar characteristics (such as the ion selectivity profile, the missing extracellular "knuckle" domain) that is likely tuned by evolution for its sensory role. The study opens a new area of research into ion channels, acid-sensing mechanism, Arthropod biology, and evolution biology. The unique features of the newly cloned channel in comparison to its homolog ASIC channels in higher species will stimulate and facilitate research into the structure-function relationship of the important yet not well-understood DEG/ENaC family of ion channels that ASIC channels belong to.

Major points to be addressed in revision:

1. The proton inhibition mechanism and the involvement of Ser376 are well-established in this study. Given the brief duration of the OFF current (about one second) in comparison to likely slow pH changes caused by environmental CO₂, it seems unlikely that the OFF current would be prominent in most real-life settings. If so, the current observed under the acidic conditions would be the depolarizing excitatory current that triggers neuronal signaling. If the authors propose the channel as a proton sensor utilizes the OFF current, the kinetic issue needs to be addressed.
2. The pore radius at the proposed upper gate, formed by Asp372, is measured at 0.86 angstrom in the presented structural model. Given that aspartate is charged at neutral pH, this places three negative charges carried by the -COO⁻ groups within tight

proximity, a situation likely containing a substantial potential energy that disfavors a closed gate. At the pH range where the channel opens, the -COO- group is likely protonated, removing the charge-charge repulsion force. If so, this energy difference would have to be compensated during activation gating. The authors should measure from their structural model and report the direct distance between the -COO- group of two aspartate side-chains, and maybe comment on how the electrical potential energy might be compensated in the closed state at neutral pH.

3. Figure 2E shows clear OFF currents upon removal of monovalent cations (and a transient current decline upon removal of Mg²⁺?). Could this phenomenon share the same mechanism with the proton OFF current? Can Ser376Asp be used to test it?

4. The authors are encouraged to comment on the potential proton-sensor of this newly identified channel, based on structural comparison to ASIC channels of higher species.

Minor points:

Line 64: "Through" should be "Guided by".

Lines 257-258: this part of the statement is unclear in its meaning; please rewrite.

Referee #2:

The manuscript by Dong et al functionally and structurally characterizes an atypical Deg/ENaC channel expressed in a set of sensory neurons of the centipede *S. subspinosus*. This channel appears to be gated by the depletion of protons from the extracellular environment. The work is robust and novel and provides structure-function insights into Deg/ENaC channels that are outside of the well characterized clades of ASICs, FaNaCs, and ENaCs. The experiments and analyses are well designed and rigorous, ranging from single cell RNA-Seq, behavioral analysis, electrophysiology, and structural determination via cryo-EM. Below, I outline some comments and concerns intended to help clarify some aspects and to improve the manuscript.

1. It seems the channel conducts robust inward currents in the absence of small external cations, presumably NMDG⁺ ions since these are the only cation present in the external solution (e.g., Figure 2E and G). Given the large size of this cation, its permeation should in the very least be noted and addressed in the manuscript. For example, why do Mg²⁺ ions, which are much smaller than NMDG⁺, not permeate while the latter does? Indeed, if this early current reflects NMDG⁺ permeation, then also please explain why there is not off response (i.e., tail current) in the presence of NMDG when transitioning from acidic to neutral pH (Figure 2E).

2. I am also curious if the authors sought to rule out whether this early current component is attributable to outward Cl⁻ permeation, or inward H⁺ permeation. Surmising protons are indeed permeating the pore, another interpretation of the observed currents is: 1) Protons elicit a small, non-desensitizing current. 2) Addition of monovalents, in the presence of protons (i.e., acidic pH conditions), causes an additive, non-desensitizing inward current mediated by these cations (Figure 2E). 3) Rapid exchange of protons for monovalents permits large, decaying/desensitizing tail currents, with the onset attributable to pore activation by protons, and the current decay to the exit of protons from the channel pore (Figure 2G).

3. Please clarify whether the current reversal experiments that were done to determine ion selectivity were conducted on sustained (non-desensitizing) currents, or tail currents elicited after proton removal. My confusion in this context, in addition to the lack of description in the results, comes from the methods section where it is unclear why, after the authors perfused various 140 mM monovalent external solutions at pH 3 (which would elicit sustained currents), the authors perfused the same solutions but at pH 7.4 (which would elicit tail currents like in Figure 2A and C). Presumably, the ramp protocols used to determine reversal potentials were run during the pH 3 perfusion, in the presence of different external monovalent cations competing with internal Na⁺ (i.e., the sustained currents?), and not the second solutions at neutral pH? Then what purpose did the pH 7.4 solution serve?

4. In addition to my point above, it would perhaps have been more informative if the authors had used voltage steps, rather than ramps, for their reversal potential experiments. This, with an invariant internal Na⁺ solution and alternating external cation and pH conditions at different step voltages, would have allowed them to determine the reversal potentials of the dual currents that result from the on/off application of extracellular protons, separately.

5. The notion that the GAS belt forms a selectivity filter (lines 149 to 150) is inconsistent with other recent works, of other Deg/ENaC channels, indicating that this motif interacts with other pore regions (i.e., the HG motif) to stabilize the pore, hence contributing to ion permeation only indirectly (e.g., Yoder & Gouaux, 2020). Specifically, in these studies the GAS belt itself is not a selectivity filter, but contributes to pore stability and hence ion permeation by stabilizing pore structures. In a related concern, since the authors only mutated the Ala383 residue in the GAS motif, and not Gly382 and Ser384, in their ion selectivity experiments (Figure S6), they cannot rule out that the other residues do not contribute to ion permeation (as stated in lines 151-155).

6. This comment relates to concerns 1 and 2 above. I am not convinced that the authors have conclusively shown that the two current components of this channel (I1 and I2) have similar ion selectivities (lines 169). Thus, it may be the first current (I1) is mediated by protons, and if so, the mutations described in the results and corresponding to Figure 4 could instead be affecting the early proton current, and not necessarily the monovalent cation currents that come after protons are removed (i.e., the tail currents).

7. I am curious why the authors did not try converting the atypical serine376 residue in the channel to glutamine, which is arguably a more canonical residue in this position present in even early-diverging Deg/ENaC homologues.

Minor Comments:

1. Please provide a more clear and detailed explanation to explain the purpose of the methyl red CO₂ experiment in the results, corresponding to figure 1 G and H and the movie EV1 (e.g., "to explore the effect of ambient CO₂ levels on the pH on small liquid droplets, emulating the aqueous lymph of sensory sensilla, we conducted experiments with...").
2. The sentence in lines 98 to 102 is unclear. How does sequence similarity of the Deg/ENaC channels suggest that the noted sensory neurons act as primary detectors of ambient mechanical and chemical stimuli? The presence of Piezo certainly suggests mechanotransduction. However, without prior knowledge of 151426 function, the logic that the sensory neurons expressing this Deg/ENaC channel are also pH sensitive needs to be laid out more clearly. For example, "since Deg/ENaC channels are often sensitive to extracellular pH, and the positioning of these sensory neurons would permit ambient pH sensing, we hypothesized that 151426 is a pH sensitive Deg/ENaC channel involved in ambient pH sensing."
3. The alignment in Figure S5 is barely legible.

Referee #3:

This study identified an ion channel in *Centropede* antenna that is activated at the end of an extracellular acidification, when the solution is changed back to the physiological pH. The channel, which belongs to the family of ENaC/degenerin channels, is named PDPNaC1. It is non-selectively permeable to monovalent cations. The authors solve the cryo-EM structure of the channel, showing that it is a trimer with similar structural organization as ASICs. With mutagenesis, the authors identify a residue that shapes ion selectivity. Since acidification induces a very small current, and the change back to the physiological pH induces a much larger transient current, it is hypothesized that protons activate the channel and block permeation at the same time. Indeed, mutation of Ser376 in the permeation pathway changes this pattern, with mutation to Asp resulting in a channel that is only activated when switching back to physiological pH, while mutation to Ala results in a channel that is strongly activated by the acidification itself. The study uses many different approaches and produces interesting findings. A few methodological aspects need to be clarified.

Main points

1. Measuring ion permeability/selectivity and ion composition of measuring solutions.
 - 1.1. I was surprised to see that the standard extracellular solution does not contain any divalent cation. Normally, recordings from mammalian cells get rapidly unstable in the absence of divalents. The absence of divalents also affects the pH dependence of channels such as ASICs, since divalents such as calcium and magnesium compete with protons for binding site. The authors need to explain in the manuscript for which reason they worked without divalents.
 - 1.2. I am also surprised that pipette solutions for ASIC recordings did not contain any calcium chelator. Is this a mistake in the methods section, or was there no chelator included?
 - 1.3. Only very few channels are permeable to Mg, however several ENaC/degenerin channels are permeable to Ca. It would make much more sense to measure the permeability to Ca, or if for some reason the authors want to avoid Ca, they could use Barium.
 - 1.4. Different ways are used here to determine the ion selectivity. In Fig.2E-F, current ratios at -80 mV are measured that show practically no difference between the monovalent ions tested. Here, the currents measured with different ions are compared to the H-induced current with NMDG. In my view, it would make more sense to take the ratio between the ion conditions (K, Cs, Mg) and Na of the same experiment, which would give a value close to 1, yielding the same conclusion, that this channel is non-selective among the monovalent ions tested. In Fig. 2I-K and for the mutants of the pore (Figs. 3 and S6), permeability ratios based on the measured reversal potentials of voltage ramps are used, which shows some monovalent ion selectivity also for the WT. This different results of the two approaches needs to be discussed. Indicate also clearly in the text whether the selectivity of the low pH-induced currents, or of the transient after switching back to the conditioning pH was measured. Indicate also the conditions of the ramp (timing, voltage range).
 - 1.5. line 153 and Figure S6, mutations of the GAS residues are mentioned, however, no data of Gly382 mutations are shown.
2. It is concluded that Gly378 contributes to ion selectivity, as stated in a subtitle. But in fact, the PDPNaC1 channel is an almost non-selective ion channel, and mutations of Gly378 render the channel more selective. Is it possible that residue Gly378 is responsible for the absence of monovalent ion selectivity in PDPNaC1? The interpretation should be adapted.
3. The interpretation that protons block channel permeation is intriguing.
 - 3.1. If Ser376 is the proton binding site, one would probably assume that the hydroxyl group of this Ser residue could be protonated or deprotonated. To my knowledge there is no example of a protein, where such a titration of Ser occurs. The authors should discuss the molecular mechanism by which they think Ser376 can bind protons.
 - 3.2. Ser376 is very close to the narrowest part of the channel pore. It is therefore highly likely that the proton block, if it occurs at the level of Ser376, depends on transmembrane voltage. In the I-V curves shown, there seemed not to be such a voltage dependence, but the range of voltages might be too small. The authors should measure currents in WT and S376 mutants at different voltages over a wider voltage range to determine whether the "proton block" indeed occurs in the pore.
4. ANAP experiments
 - 4.1. From Fig. 4D it is difficult to really see how much buried Ser376 is in the structure. Would it be possible to show a view from

top, as in Fig. 3I for other residues, to better visualize the orientation?

4.2. It is shown that the currents of Ser376ANAP are transient and that at the acidic pH, the current is very small. The change in ANAP emission spectrum is therefore measured in a situation with very low conductance. Would it also be possible to measure the ANAP fluorescence during the short time after switching back to the conditioning solution when the big current is measured? Indicate for the measurement of the emission spectrum, what solution changes were made, and what the timing was.

4.3. For Fig. 4F and G, indicate the pH at which the black trace was recorded.

4.4. To my knowledge, the intrinsic pH dependence of ANAP has so far not been measured. There is a risk that pH3 affects the intrinsic ANAP emission, independently of its exposure. Could you carry out any controls to exclude this possibility?

4.5. In the methods it is indicated that the cells were incubated with ANAP. Since ANAP does not easily cross membranes, would it be possible that the ANAP methyl ester was used instead?

Specific points

1. PAC-ko HEK293 cells were used. Many HEK293 cell types also express endogenous ASIC1a. Can the authors confirm in the manuscript that their cell line did not express any endogenous ASICs?

2. Fig. 1E, colocalization with Piezo2, a quantitative analysis of the colocalization should be provided.

3. Line 268, provide the supplier of the centipedes, not only the region.

4. Line 314, Western blots, specify the types and concentrations of protease and phosphate inhibitors.

5. Line 513, please check the equation. It seems wrong, unless you use negative Hill coefficients.

6. Line 570, ref. Chen et al is incomplete

7. Indicate in all the protocols also the pH between acidic stimulations.

8. For the western blots in Fig. S8, a control from non-transfected cells needs to be included.

Dear Dr. Martina,

Thank you for considering our manuscript entitled "*A Proton-Gated Channel Identified in the Centipede Antenna*" (Tracking #: EMBOR-2025-62006-T), and for providing us with the opportunity to submit a revised version.

We are sincerely grateful to the reviewers for their insightful comments. We have carefully addressed all concerns raised and revised the manuscript accordingly. All changes are clearly marked using track changes.

Below, we provide a detailed, point-by-point response to each of the reviewers' comments. We believe that the additional experiments and revised explanations have strengthened the manuscript and clarified our findings.

Please do not hesitate to let us know if any further revisions are required. We appreciate your time and consideration, and we look forward to hearing from you.

Best regards,

Yunfei Wang, Ph.D. Professor

College of Wildlife and Protected Area, Northeast Forestry University, Harbin 150040, China

Tel. +86-17787292103; e-mail: wangyunfei2020@nefu.edu.cn

Referee #1:

In this study, Dong and colleagues cloned an apparent ASIC homolog in centipede, solved its structure with cryo-EM, characterized its function and delineated the structure-function correlates, and proposed that the channel serves as a CO₂ sensor. Through this effort, they identified the channel's activation gate, the ion selectivity filter, and a proton binding site that gives the channel its prominent proton-dependent conductance. This is a large body of work that presents an interesting new ion channel with some peculiar characteristics (such as the ion selectivity profile, the missing extracellular "knuckle" domain) that is likely tuned by evolution for its sensory role. The study opens a new area of research into ion channels, acid-sensing mechanism, Arthropod biology, and evolution biology. The unique features of the newly cloned channel in comparison to its homolog ASIC channels in higher species will stimulate and facilitate research into the structure-function relationship of the important yet not well-understood DEG/ENaC family of ion channels that ASIC channels belong to.

Response: We sincerely thank the reviewer for the thoughtful and encouraging comments. We are pleased that the significance, novelty, and potential impact of our study were recognized. The unique features of PDPNaC1 indeed provide exciting new insights into acid-sensing mechanisms and the DEG/ENaC family. We greatly appreciate your positive evaluation.

Major points to be addressed in revision:

1. The proton inhibition mechanism and the involvement of Ser376 are well-established in this study. Given the brief duration of the OFF current (about one second) in comparison to likely slow pH changes caused by environmental CO₂, it seems unlikely that the OFF current would be prominent in most real-life settings. If so, the current observed under the acidic conditions would be the depolarizing excitatory current that triggers neuronal signaling. If the authors propose the channel as a proton sensor utilizes the OFF current, the kinetic issue needs to be addressed.

Response: We appreciate your insightful comment regarding the physiological relevance of the OFF current (I₂) and its kinetics. While we acknowledge the brief duration of I₂ (~1 second), we propose that this transient current mediates rapid neuronal signaling triggered by fast pH fluctuations in the microenvironment, which are indeed possible in physiological contexts.

Our experiments demonstrate that both the introduction and removal of CO₂ induce significant pH changes in droplets (Fig. 1H and I). Notably, smaller droplets exhibit faster pH response kinetics: in our assays, the smallest detectable droplet ($\varnothing = 630 \mu\text{m}$) showed a pH recovery time of ~9 seconds following CO₂ withdrawal (Fig. 1I). However, centipede antennal sensilla are substantially smaller, with diameters of less than 30 μm (Fig. 1G). Given the inverse correlation between droplet size and pH recovery time, we predict that pH transitions in these microstructures could occur on the millisecond scale, aligning well with the rapid kinetics of I₂ activation.

Moreover, in natural settings, respiration from mammalian predators can generate sharp local CO₂ fluctuations, potentially causing rapid pH oscillations within the sensillar environment. The I₂ response of PDPNaC1 may thus function as a rapid alarm

mechanism to detect such environmental threats. Additionally, the active, high-frequency sweeping of the antennae (see Movie EV2) likely facilitates fast transitions in chemical exposure, further enhancing the temporal alignment between environmental stimuli and the channel's response kinetics.

2. The pore radius at the proposed upper gate, formed by Asp372, is measured at 0.86 angstrom in the presented structural model. Given that aspartate is charged at neutral pH, this places three negative charges carried by the -COO- groups within tight proximity, a situation likely containing a substantial potential energy that disfavors a closed gate. At the pH range where the channel opens, the -COO- group is likely protonated, removing the charge-charge repulsion force. If so, this energy difference would have to be compensated during activation gating. The authors should measure from their structural model and report the direct distance between the -COO- group of two aspartate side-chains, and maybe comment on how the electrical potential energy might be compensated in the closed state at neutral pH.

Response: Thank you for this thoughtful and technically insightful question. In our closed-state structure, the distance between the carbonyl oxygen atoms of the three Asp372 side chains is approximately 3.6 Å. At this close range, electrostatic repulsion among the negatively charged -COO⁻ groups are indeed plausible and could contribute to an energetically unfavorable conformation.

To mitigate this repulsion, we propose that nearby Ser368 residues, which are positioned in close proximity, may provide stabilizing interactions through their hydroxyl groups. These side chains are well-positioned to form hydrogen bonds with the carboxylate oxygens of Asp372, thereby helping to locally stabilize the closed conformation despite the charge repulsion.

Furthermore, studies on ASIC channels have demonstrated that the equivalent residue, Asp433, is protonated under activating conditions, which reduces electrostatic repulsion and facilitates pore opening. By analogy, we propose that Asp372 in PDPNaC1 may undergo similar protonation during activation, thereby contributing to a more favorable energetic landscape for pore opening. This protonation-driven transition would relieve electrostatic strain in the closed state and may serve as a gating mechanism coupled to local pH changes.

3. Figure 2E shows clear OFF currents upon removal of monovalent cations (and a

transient current decline upon removal of Mg²⁺?). Could this phenomenon share the same mechanism with the proton OFF current? Can Ser376Asp be used to test it?

Response: Thank you for highlighting this important experimental observation. The OFF currents initially observed upon the removal of monovalent cations in Fig. 2E were artifacts caused by a brief, unintended exposure to pH 7.4 during solution exchange, which led to transient proton dissociation and mimicked an OFF current. We have since repeated the experiment using an improved perfusion protocol that eliminates transient pH fluctuations. As shown in the updated Fig. 2E, no discernible OFF current is observed under these conditions, confirming that the previous response was not directly caused by monovalent ion removal.

While Ser376Asp is a valuable mutant for probing proton interactions, it is not applicable in this case, as the transient current resulted entirely from a technical artifact rather than a physiological mechanism.

4. The authors are encouraged to comment on the potential proton-sensor of this newly identified channel, based on structural comparison to ASIC channels of higher species.

Response: Thank you for the helpful suggestion. As recommended, we have added a discussion on the potential proton-sensing mechanism of PDPNaC1, based on structural comparisons with ASIC1a from higher species. Specifically, we highlight both conserved and divergent residues in the extracellular domain that may underlie distinct modes of proton recognition and gating. The revised text can be found in the Discussion section (lines 255-263).

Minor points:

Line 64: "Through" should be "Guided by".

Response: Thank you for the suggestion. The text has been corrected accordingly.

Lines 257-258: this part of the statement is in its meaning; please rewrite.

Response: Thank you for pointing this out. We have revised the sentence as follows: "Consequently, the neuronal localization and gating kinetics of PDPNaC1 likely work in synergy with antennal movements to detect pH fluctuations—such as those caused by predator respiration—in natural environments. This may confer an adaptive advantage to centipedes by enhancing their ability to sense transient acidic conditions." (lines 317-321)

Referee #2:

The manuscript by Dong et al functionally and structurally characterizes an atypical Deg/ENaC channel expressed in a set of sensory neurons of the centipede *S. subspinipes*. This channel appears to be gated by the depletion of protons from the extracellular environment. The work is robust and novel and provides structure-function insights into Deg/ENaC channels that are outside of the well characterized clades of ASICs, FaNaCs, and ENaCs. The experiments and analyses are well designed

and rigorous, ranging from single cell RNA-Seq, behavioral analysis, electrophysiology, and structural determination via cryo-EM. Below, I outline some comments and concerns intended to help clarify some aspects and to improve the manuscript.

Response: Thank you very much for your thorough reading and constructive comments. We greatly appreciate your positive evaluation of our work. Detailed point-by-point responses to your comments are provided below. We believe that the revisions made in response to your suggestions have substantially improved the clarity and overall quality of the manuscript.

1. It seems the channel conducts robust inward currents in the absence of small external cations, presumably NMDG⁺ ions since these are the only cation present in the external solution (e.g., Figure 2E and G). Given the large size of this cation, its permeation should in the very least be noted and addressed in the manuscript. For example, why do Mg²⁺ ions, which are much smaller than NMDG⁺, not permeate while the latter does? Indeed, if this early current reflects NMDG⁺ permeation, then also please explain why there is not off response (i.e., tail current) in the presence of NMDG when transitioning from acidic to neutral pH (Figure 2E).

Response: Thank you for raising this important point. The inward currents observed at pH 3.0 in the presence of NMDG-Cl are indeed mediated by H⁺ permeation rather than by NMDG⁺ influx. We apologize for any confusion caused by insufficient clarity in the original figure legend and text.

To further support this conclusion, we conducted additional experiments using [EMIM][MeSO₃] (1-ethyl-3-methylimidazolium methanesulfonate), a salt composed of large, impermeant cations (EMIM⁺) and anions (MeSO₃⁻), in both extracellular and intracellular solutions. Under these conditions, robust acid-activated inward currents were still observed, strongly indicating that protons, not organic cations, are the charge carriers (see revised figure below).

As also noted by Reviewer #1, the absence of a tail current when transitioning from acidic (pH 3.0) to neutral pH in the presence of NMDG-Cl is consistent with NMDG⁺ and Cl⁻ being impermeant, further supporting our interpretation (updated Fig. 2E). We have revised the relevant description in the Results section to clarify this point (Lines 134-138).

Experimental conditions for reference:

Both intracellular and extracellular solutions contained 140 mM [EMIM][MeSO₃] and 10 mM HEPES, with osmolality adjusted to 280-300 mOsm/kg using glucose. pH adjustments were made using methanesulfonic acid and 1-ethyl-3-methylimidazolium hydroxide.

2. I am also curious if the authors sought to rule out whether this early current component is attributable to outward Cl⁻ permeation, or inward H⁺ permeation. Surmising protons are indeed permeating the pore, another interpretation of the observed currents is: 1) Protons elicit a small, non-desensitizing current. 2) Addition of monovalents, in the presence of protons (i.e., acidic pH conditions), causes an additive, non-desensitizing inward current mediated by these cations (Figure 2E). 3) Rapid exchange of protons for monovalents permits large, decaying/desensitizing tail currents, with the onset attributable to pore activation by protons, and the current decay to the exit of protons from the channel pore (Figure 2G).

Response: Thank you for your thoughtful analysis of the current components shown in Figures 2E and 2G. As noted in our response to your Comment#1, we have ruled out outward Cl⁻ permeation and confirmed that the early current component is mediated by inward H⁺ permeation in the presence of NMDG-Cl. This conclusion is further supported by experiments using impermeant ions ([EMIM][MeSO₃]), as described previously.

Regarding the tail currents observed in Figure 2G, we have confirmed that Na⁺, K⁺, and Cs⁺ serve as the permeating ions under these conditions. Based on our data, we propose the following mechanism:

1. The initial H⁺ evoked channel open induces a proton block at the Ser376 residue, maintaining the channel in a non-conducting or weakly conducting state.
2. Upon washout of extracellular protons (i.e., at neutral pH), proton dissociation relieves this block, allowing the pore to transiently conduct Na⁺, K⁺, or Cs⁺-resulting in a rapid, desensitizing tail current (I₂).
3. The decay phase of I₂ reflects the transition of the channel from an open to a closed state following the loss of proton gating.

We have revised the Results (Lines 133-141) sections accordingly to clarify this interpretation.

3. Please clarify whether the current reversal experiments that were done to determine ion selectivity were conducted on sustained (non-desensitizing) currents, or tail currents elicited after proton removal. My confusion in this context, in addition to the lack of description in the results, comes from the methods section where it is unclear why, after the authors perfused various 140 mM monovalent external solutions at pH 3 (which would elicit sustained currents), the authors perfused the same solutions but at pH 7.4 (which would elicit tail currents like in Figure 2A and C). Presumably, the ramp protocols used to determine reversal potentials were run during the pH 3 perfusion, in the presence of different external monovalent cations competing with internal Na⁺ (i.e., the sustained currents?), and not the second solutions at neutral pH? Then what purpose did the pH 7.4 solution serve?

Response: Thank you for your careful reading and for raising this critical point. We apologize for the lack of clarity in the original Methods section and figure legends regarding the conditions used for ion selectivity measurements.

To clarify, the current reversal experiments were conducted using the tail currents (I₂)

elicited upon proton removal—specifically, by switching from pH 3.0 to pH 7.4 in the presence of the same external monovalent cation. We have now revised the corresponding sections in the Results (Lines 141-145), Methods (Lines 534-541), and Figure legends (Lines 933-936).

Both I_1 (the sustained current at pH 3.0) and I_2 (the tail current at pH 7.4) exhibit similar ion selectivity—permeable to monovalent cations but impermeable to divalent cations (Fig. 2E-H). However, I_1 is recorded under strongly acidic conditions and is therefore subject to proton block and influenced by the proton electrochemical gradient, which could confound accurate assessment of reversal potentials. To minimize these confounding factors and obtain more reliable selectivity data, we performed the primary reversal potential measurements using I_2 , under conditions in which the proton gradient is eliminated and the signal remains robust.

To further address your concern, we additionally performed reversal potential measurements using I_1 at pH 3.0. These results revealed an ion selectivity trend consistent with that observed for I_2 (see figure below), thereby validating our original conclusions.

(A) Representative I-V traces recorded at pH 3.0 with 140 mM NaCl in the pipette and equimolar extracellular monovalent cations. (B) Relative ion permeabilities for PDPNaC1 at pH 3.0 ($n = 5$ cells per condition).

4. In addition to my point above, it would perhaps have been more informative if the authors had used voltage steps, rather than ramps, for their reversal potential experiments. This, with an invariant internal Na^+ solution and alternating external cation and pH conditions at different step voltages, would have allowed them to determine the reversal potentials of the dual currents that result from the on/off application of extracellular protons, separately.

Response: Thank you very much for this constructive suggestion. Following your recommendation, we performed additional experiments using voltage step protocols to independently assess the reversal potentials of the dual currents elicited by the application and removal of extracellular protons.

As shown in the figure below, the reversal potentials determined using voltage steps are highly consistent with those obtained from our original voltage ramp recordings. This agreement further supports the reliability of our ion selectivity measurements and validates our interpretation of the dual current components.

(A) Representative I-V relationships of PDPNaC1 recorded using voltage step protocols at pH 3.0, with 140 mM NaCl in the pipette and equimolar extracellular monovalent cations.

(B) Relative ion permeabilities of PDPNaC1 at pH 3.0, determined using both voltage step and ramp protocols (n = 5 cells per group).

(C) Representative I-V relationships of PDPNaC1 I_2 currents recorded during pH transition (from pH 3.0 to 7.4) at holding potentials of -40, -20, +20, or +40 mV, using 140 mM NaCl in the pipette and equimolar extracellular monovalent cations (n = 5 cells per condition).

(D) Relative ion permeabilities of PDPNaC1 during the pH transition, determined using both voltage step and ramp protocols (n = 5 cells per group).

5. The notion that the GAS belt forms a selectivity filter (lines 149 to 150) is inconsistent with other recent works, of other Deg/ENaC channels, indicating that this motif interacts with other pore regions (i.e., the HG motif) to stabilize the pore, hence contributing to ion permeation only indirectly (e.g., Yoder & Gouaux, 2020). Specifically, in these studies the GAS belt itself is not a selectivity filter, but contributes to pore stability and hence ion permeation by stabilizing pore structures. In a related concern, since the authors only mutated the Ala383 residue in the GAS motif, and not Gly382 and Ser384, in their ion selectivity experiments (Figure S6), they cannot rule out that the other residues do not contribute to ion permeation (as stated in lines 151-155).

Response: Thank you for your insightful comment. We agree that in several other DEG/ENaC family members, including those studied by Yoder & Gouaux (2020), the GAS motif has been shown to contribute to ion permeation indirectly—primarily by structurally stabilizing the pore through interactions with motifs such as HG, rather than acting as the ion selectivity filter itself.

In the case of PDPNaC1, this interpretation is entirely consistent with our findings. Importantly, PDPNaC1 lacks the canonical HG motif, suggesting a distinct ion conduction architecture. More importantly, our mutagenesis data demonstrate that none of the three residues in the GAS belt—Gly382, Ala383, or Ser384—affect ion selectivity. The results for Ser384 were included in the original Appendix Fig. S6, while the data for Gly382 were inadvertently omitted and have now been incorporated into the revised Appendix Fig. S7.

Together, these results support the view that, in PDPNaC1, the GAS motif does not serve as the selectivity filter but may instead contribute to the overall pore architecture or structural stability.

6. This comment relates to concerns 1 and 2 above. I am not convinced that the authors

have conclusively shown that the two current components of this channel (I₁ and I₂) have similar ion selectivities (lines 169). Thus, it may be the first current (I₁) is mediated by protons, and if so, the mutations described in the results and corresponding to Figure 4 could instead be affecting the early proton current, and not necessarily the monovalent cation currents that come after protons are removed (i.e., the tail currents).

Response: Thank you for this thoughtful comment. As noted in our responses to Concerns 3 and 4, our data support that both I₁ (the sustained current at acidic pH) and I₂ (the tail current upon proton removal) exhibit similar ion selectivity, being permeable to monovalent cations but not to divalent cations. These findings suggest that I₁ and I₂ represent different ion conduction phases of the same channel state, rather than distinct channel states (i.e., activated state and desensitized state).

Regarding the effects of mutations shown in Figure 4, we agree that the apparent differences in the I₁ component among S376 mutants are likely due to altered proton-binding affinities, which in turn modulate the extent of proton-mediated inhibition. Specifically, in the S376D mutant, stronger proton binding leads to more pronounced inhibition during the I₁ phase, while in the S376A mutant, weaker binding results in milder inhibition (Fig. 4B, C). In contrast, since I₂ occurs after proton removal, it is no longer influenced by proton occupancy at the binding site, and thus the current amplitude remains comparable across different mutants (Fig. 4B, C).

We also thank Reviewer #3 for highlighting the voltage knock-off experiments, which further support this “proton block” interpretation (updated in Fig. 4D).

7. I am curious why the authors did not try converting the atypical serine376 residue in the channel to glutamine, which is arguably a more canonical residue in this position present in even early-diverging Deg/ENaC homologues.

Response: Thank you for this thoughtful suggestion. We have indeed tested the S376Q mutant in our experiments and found that both I₁ and I₂ currents are completely abolished.

One possible explanation is that the glutamine substitution mimics a protonated aspartate residue, thereby producing persistent proton-like inhibition of the channel. Alternatively, the S376Q mutation may disrupt local structural integrity, leading to a non-functional channel conformation. At present, it is difficult to definitively distinguish between these two possibilities.

Representative current trace of the S376Q mutant recorded at -80 mV under different pH conditions.

Minor Comments:

1. Please provide a more clear and detailed explanation to explain the purpose of the

methyl red CO₂ experiment in the results, corresponding to figure 1 G and H and the movie EV1 (e.g., "to explore the effect of ambient CO₂ levels on the pH on small liquid droplets, emulating the aqueous lymph of sensory sensilla, we conducted experiments with...").

Response: Thank you for your careful suggestion. We have revised the relevant text to more clearly explain the purpose of the methyl red — CO₂ experiment, as follows:

“The antennal surface of centipedes is adorned with various types of sensilla (Fig. 1G), which are small cuticular structures likely filled with lymph-like body fluid. These microstructures may undergo rapid chemical changes in response to environmental stimuli, such as pH fluctuations induced by ambient CO₂. To emulate this process, we conducted experiments using droplets of varying diameters. Upon application and removal of CO₂, we observed rapid acidification and pH recovery within seconds. Notably, the rate of pH change was inversely correlated with droplet size (Fig. 1H and I; Movie EV1).”

2. The sentence in lines 98 to 102 is unclear. How does sequence similarity of the Deg/ENaC channels suggest that the noted sensory neurons act as primary detectors of ambient mechanical and chemical stimuli? The presence of Piezo certainly suggests mechanotransduction. However, without prior knowledge of 151426 function, the logic that the sensory neurons expressing this Deg/ENaC channel are also pH sensitive needs to be laid out more clearly. For example, "since Deg/ENaC channels are often sensitive to extracellular pH, and the positioning of these sensory neurons would permit ambient pH sensing, we hypothesized that 151426 is a pH sensitive Deg/ENaC channel involved in ambient pH sensing."

Response: Thank you for this helpful comment. We have revised the original text as follows to improve the clarity and logical flow.

“DEG/ENaC channels are frequently sensitive to extracellular pH, prompting the hypothesis that isoform 151426 may function as an acid-sensitive receptor. In addition, this isoform is selectively expressed in antennal sensory neurons located at the surface of the antenna, a position well-suited for direct exposure to ambient chemical cues. Taken together, these features suggest that neurons expressing 151426 may act as primary sensors of environmental pH, potentially responding to acidic stimuli such as CO₂ exposure.”

3. The alignment in Figure S5 is barely legible.

Response: Thank you for pointing this out. We have improved the resolution in the updated Appendix Fig. S6 to ensure that the sequence alignment is clearly legible and the conserved motifs are easily distinguishable.

Referee #3:

This study identified an ion channel in Centripede antenna that is activated at the end of an extracellular acidification, when the solution is changed back to the physiological pH. The channel, which belongs to the family of ENaC/degenerin channels, is named

PDPNaCl. It is non-selectively permeable to monovalent cations. The authors solve the cryo-EM structure of the channel, showing that it is a trimer with similar structural organization as ASICs. With mutagenesis, the authors identify a residue that shapes ion selectivity. Since acidification induces a very small current, and the change back to the physiological pH induces a much larger transient current, it is hypothesized that protons activate the channel and block permeation at the same time. Indeed, mutation of Ser376 in the permeation pathway changes this pattern, with mutation to Asp resulting in a channel that is only activated when switching back to physiological pH, while mutation to Ala results in a channel that is strongly activated by the acidification itself. The study uses many different approaches and produces interesting findings. A few methodological aspects need to be clarified.

Response: We sincerely thank the reviewer for the thoughtful evaluation of our work and for highlighting the novelty and significance of our findings on the PDPNaCl ion channel in centipede antennae. We greatly appreciate your recognition of the multidisciplinary approach and key insights presented in our study.

Regarding the methodological points raised, we have carefully addressed each of them in the detailed responses below. We have also revised the manuscript accordingly to clarify these aspects and improve the overall presentation of the work.

Main points

1. Measuring ion permeability/selectivity and ion composition of measuring solutions.
1.1. I was surprised to see that the standard extracellular solution does not contain any divalent cation. Normally, recordings from mammalian cells get rapidly unstable in the absence of divalents. The absence of divalents also affects the pH dependence of channels such as ASICs, since divalents such as calcium and magnesium compete with protons for binding site. The authors need to explain in the manuscript for which reason they worked without divalents.

Response: Thank you for this important observation. In our preliminary experiments, we compared extracellular solutions with and without divalent cations and found that PDPNaCl current responses to both proton application and washout were comparable under the two conditions (see figure below). This suggests that divalent cations do not significantly influence the channel's proton sensitivity or gating behavior.

Moreover, as PDPNaCl is impermeable to divalent cations such as Ca^{2+} and Mg^{2+} , we opted to exclude them from the standard extracellular solution in order to simplify the ionic environment and facilitate interpretation of monovalent cation selectivity. We have now clarified this choice in the revised Methods section (Lines 517-521).

Representative current traces of PDPNaCl recorded at -80 mV under different pH

conditions, in the presence or absence of extracellular Ca^{2+} . Recordings were performed using 140 mM NaCl in the pipette solution.

1.2. I am also surprised that pipette solutions for ASIC recordings did not contain any calcium chelator. Is this a mistake in the methods section, or was there no chelator included?

Response: We thank the reviewer for this helpful observation. There was indeed an oversight in the original Methods section. The pipette solution used for ASIC recordings did include a calcium chelator, and we have now corrected this in the revised version of the manuscript to reflect the experimental conditions (Line 543) accurately.

1.3. Only very few channels are permeable to Mg, however several ENaC/degenerin channels are permeable to Ca. It would make much more sense to measure the permeability to Ca, or if for some reason the authors want to avoid Ca, they could use Barium.

Response: Thank you for this insightful comment. In our experiments, PDPNaCl exhibited no measurable permeability to divalent cations, including Mg^{2+} , Ca^{2+} , and Ba^{2+} (see figure below). The inclusion of Mg^{2+} in the main figure was intended as a representative example to illustrate the lack of divalent cation permeability, rather than a specific exclusion of Ca^{2+} or Ba^{2+} . For clarity, we have now included the permeability data for Ca^{2+} and Ba^{2+} in the revised Fig. 2F and 2H.

(A) Representative current traces of PDPNaCl with 110 mM of each external divalent cation and 140 mM NMDG⁺ in the pipette (V_m = -80 mV). (B) Representative current traces of PDPNaCl recorded from -80 to +80 mV under different pH conditions. Recordings were performed using 140 mM NMDG-Cl (pH 7.4) in the pipette and various extracellular solutions, which were sequentially switched to: 140 mM NMDG-Cl (pH 3.0), 110 mM MgCl₂/CaCl₂/BaCl₂ (pH 7.4), 110 mM MgCl₂/CaCl₂/BaCl₂ (pH 3.0), and 110 mM MgCl₂/CaCl₂/BaCl₂ (pH 7.4).

1.4. Different ways are used here to determine the ion selectivity. In Fig. 2E-F, current ratios at -80 mV are measured that show practically no difference between the monovalent ions tested. Here, the currents measured with different ions are compared to the H-induced current with NMDG. In my view, it would make more sense to take the ratio between the ion conditions (K, Cs, Mg) and Na of the same experiment, which would give a value close to 1, yielding the same conclusion, that this channel is non-

selective among the monovalent ions tested. In Fig. 2I-K and for the mutants of the pore (Figs. 3 and S6), permeability ratios based on the measured reversal potentials of voltage ramps are used, which shows some monovalent ion selectivity also for the WT. This different results of the two approaches needs to be discussed. Indicate also clearly in the text whether the selectivity of the low pH-induced currents, or of the transient after switching back to the conditioning pH was measured. Indicate also the conditions of the ramp (timing, voltage range).

Response: We thank the reviewer for raising this important point regarding the assessment of ion selectivity using peak current measurements versus reversal potential analysis. These two approaches are based on distinct electrophysiological principles and serve different experimental purposes, which may account for the observed differences in the results.

In our study, peak current amplitudes at -80 mV were used as a qualitative indicator to assess whether PDPNaC1 conducts various monovalent cations. However, current amplitude is influenced not only by ion permeability but also by the electrochemical driving force and channel open probability. In our setup—with 140 mM of each external monovalent cation and 140 mM NMDG⁺ in the pipette—a strong inward driving force may have saturated the currents, thereby masking subtle differences in ion permeability. By contrast, reversal potential measurements obtained from voltage ramp protocols provide a more quantitative and robust evaluation of relative ion permeability under bi-ionic conditions, as they are based on the zero-current point and are unaffected by driving force or channel gating kinetics.

In Fig. 2F, we used the H⁺-induced current (recorded under NMDG⁺ background) as a stable reference to qualitatively compare relative conductance among different monovalent cations. However, we agree with the reviewer that for an accurate, quantitative assessment, it is more appropriate to normalize against Na⁺. Accordingly, in our reversal potential-based measurements (Figs. 2I-K and 3, and Appendix Fig. S7), Na⁺ was used as the standard for calculating relative permeability ratios.

To improve clarity, we have now explicitly stated in the figure legends that reversal potentials were derived from tail currents (I₂) elicited upon switching extracellular pH from 3.0 to 7.4 (Lines 933-936). In addition, the detailed parameters of the voltage ramp protocol—including timing and voltage range—have been added to the revised Methods section (Lines 539-541, 547-548).

1.5. line 153 and Figure S6, mutations of the GAS residues are mentioned, however, no data of Gly382 mutations are shown.

Response: Thank you for pointing this out. We also thank Reviewer #2 for raising the same concern. The ion selectivity data for the Gly382 mutants were unintentionally omitted in the original submission and have now been included in the updated Appendix Fig. S7.

2. It is concluded that Gly378 contributes to ion selectivity, as stated in a subtitle. But in fact, the PDPNaC1 channel is an almost non-selective ion channel, and mutations of Gly378 render the channel more selective. Is it possible that residue Gly378 is

responsible for the absence of monovalent ion selectivity in PDPNaC1? The interpretation should be adapted.

Response: Thank you very much for this insightful suggestion. We agree that the original subtitle may have unintentionally implied that Gly378 determines ion selectivity in a classical sense. To better reflect our findings, we have revised the subtitle to: “Gly378 mutations enhance ion selectivity in PDPNaC1.” We have also updated the corresponding text in the main body (Lines 173, 185-187).

3. The interpretation that protons block channel permeation is intriguing.

3.1. If Ser376 is the proton binding site, one would probably assume that the hydroxyl group of this Ser residue could be protonated or deprotonated. To my knowledge there is no example of a protein, where such a titration of Ser occurs. The authors should discuss the molecular mechanism by which they think Ser376 can bind protons.

Response: Thank you for raising this critical point. We agree that direct protonation or deprotonation of a serine hydroxyl group is highly unlikely under physiological conditions due to its high pKa. However, we propose that Ser376 contributes to proton sensitivity through an indirect mechanism, by forming hydrogen bonds with hydrated proton species in the pore environment.

This concept has been supported by prior computational studies of synthetic proton channels such as LS2, in which serine side chains were shown to form strong, long-lived hydrogen bonds with water molecules in the first solvation shell of protonated species—specifically Zundel (H_5O_2^+) and Eigen (H_9O_4^+) cations (Wu and Voth, *Biophys. J.*, 2003). Within the specific microenvironment of the PDPNaC1 pore, Ser376 may interact with such hydrated protons via its hydroxyl group, thereby indirectly contributing to a proton-blocking effect.

Importantly, this proposed mechanism is also supported by our new electrophysiological data (presented in response to Comment 3.2 and in updated Fig. 4D), which show that proton block at Ser376 is voltage-dependent—a hallmark of interactions occurring within the membrane electric field. In particular, we found that the I_1/I_2 ratio increases at more negative voltages in wild-type and Ser376Cys channels, consistent with relief of proton block under stronger driving forces. These functional data reinforce the interpretation that Ser376 mediates a proton-sensitive interaction within the pore.

We have now incorporated both the mechanistic explanation and the supporting functional data into the revised Discussion (Lines 289-293) section.

3.2. Ser376 is very close to the narrowest part of the channel pore. It is therefore highly likely that the proton block, if it occurs at the level of Ser376, depends on transmembrane voltage. In the I-V curves shown, there seemed not to be such a voltage dependence, but the range of voltages might be too small. The authors should measure currents in WT and S376 mutants at different voltages over a wider voltage range to determine whether the "proton block" indeed occurs in the pore.

Response: Thank you very much for this constructive suggestion. In response, we performed additional recordings to examine the voltage dependence of proton block in

both wild-type PDPNaC1 and Ser376 mutants across an extended voltage range (updated Fig. 4D).

We found that the I_1/I_2 ratio increased at more negative membrane potentials in both wild-type and Ser376Cys channels, indicating that proton block is progressively relieved under stronger hyperpolarizing driving forces. In contrast, the Ser376Ala mutant exhibited no voltage dependence, consistent with the near-complete loss of proton-mediated inhibition. Notably, in the Ser376Asp mutant, the I_1 component was abolished across all tested voltages, likely due to strong, constitutive proton binding and persistent block.

These results support the conclusion that proton block in PDPNaC1 occurs within the transmembrane electric field, consistent with a pore-localized mechanism involving Ser376. This voltage sensitivity further reinforces the role of Ser376 in mediating proton interaction. We have incorporated these findings into the revised manuscript (Lines 213-223).

4. ANAP experiments

4.1. From Fig. 4D it is difficult to really see how much buried Ser376 is in the structure. Would it be possible to show a view from top, as in Fig. 3I for other residues, to better visualize the orientation?

Response: Thank you for this helpful suggestion. To better illustrate the spatial positioning of Ser376, we have now included a top-view representation of its side chain in the updated Fig. 5A. This revised view more clearly shows the residue's buried orientation within the channel pore.

4.2. It is shown that the currents of Ser376ANAP are transient and that at the acidic pH, the current is very small. The change in ANAP emission spectrum is therefore measured in a situation with very low conductance. Would it also be possible to measure the ANAP fluorescence during the short time after switching back to the conditioning solution when the big current is measured? Indicate for the measurement of the emission spectrum, what solution changes were made, and what the timing was.

Response: Thank you for this helpful suggestion. In the revised manuscript, we performed ANAP fluorescence measurements immediately after switching the extracellular solution from pH 3.0 back to pH 7.4, corresponding to the time window during which the prominent tail current (I_2) is observed. Under these conditions, we detected an ~8 nm red shift in the ANAP emission spectrum compared to the resting closed state at pH 7.4, suggesting that the side chain of Ser376 remains in an exposed conformation during this transient, conductive phase.

We have updated the corresponding figure and legend (revised Fig. 5B-F) to clearly indicate the solution change protocol and the precise timing of the emission spectrum acquisition.

4.3. For Fig. 4F and G, indicate the pH at which the black trace was recorded.

Response: Thank you for pointing this out. We have now clearly indicated the pH conditions corresponding to each ANAP fluorescence trace, including the black trace,

in the revised figure legend (Lines 997-999).

4.4. To my knowledge, the intrinsic pH dependence of ANAP has so far not been measured. There is a risk that pH3 affects the intrinsic ANAP emission, independently of its exposure. Could you carry out any controls to exclude this possibility?

Response: Thank you for raising this important point. To assess the potential intrinsic pH sensitivity of ANAP, we measured the emission spectra of free ANAP in solution across a range of pH conditions (updated Appendix Fig. S9). The results show that ANAP exhibits a blue shift of approximately 10 nm as the pH decreases from 7.0 to 3.0. Importantly, this intrinsic blue shift is in the opposite direction to the red shift we observed for Ser376ANAP upon transitioning from pH 7.4 to pH 3.0. This indicates that the spectral change seen in the channel context is not attributable to ANAP's inherent pH responsiveness, but rather reflects a buried-to-exposed conformational transition of the Ser376 side chain within the protein environment.

These findings suggest that the microenvironmental change at Ser376 is likely even more pronounced than indicated by the raw fluorescence shift, further strengthening our interpretation. We have incorporated these findings into the revised manuscript (Lines 237-240).

4.5. In the methods it is indicated that the cells were incubated with ANAP. Since ANAP does not easily cross membranes, would it be possible that the ANAP methyl ester was used instead?

Response: We appreciate the reviewer's comment. In our experiments, we used ANAP itself (not the methyl ester form), and found that it could be efficiently incorporated into cells when co-applied with Lipofectamine 3000 during the transfection process. Under these conditions, ANAP was successfully delivered into cells and incorporated into the target channel proteins, resulting in robust and position-specific fluorescence signals. This approach has been validated in our previous studies, where ANAP was similarly used to monitor conformational changes in ion channels (Lu et al., *PNAS*, 2022; Yang et al., *PNAS*, 2020). We have clarified this point in the revised Methods section to avoid confusion (Lines 621).

Specific points

1. PAC-ko HEK293 cells were used. Many HEK293 cell types also express endogenous ASIC1a. Can the authors confirm in the manuscript that their cell line did not express any endogenous ASICs?

Response: We acknowledge that certain HEK293 cell lines can exhibit low levels of endogenous ASIC expression. In our study, however, endogenous ASIC-like currents were minimal under our recording conditions and were not consistently observed across cells (Appendix Fig. S12). Moreover, ASIC channels are known to undergo rapid desensitization at low pH, further reducing their potential contribution. Based on these observations, we consider the influence of endogenous ASICs on PDPNaC1 current recordings to be negligible. This clarification has been added to the revised manuscript (Lines 503-506).

2. Fig. 1E, colocalization with Piezo2, a quantitative analysis of the colocalization should be provided.

Response: Thank you for your valuable suggestion. We have now included a quantitative analysis of the colocalization, which is presented in the updated Fig. 1F in the revised manuscript.

3. Line 268, provide the supplier of the centipedes, not only the region.

Response: Thank you for pointing this out. We have now included the name of the supplier of the centipedes, in addition to the region, in the revised manuscript (Lines 330-331).

4. Line 314, Western blots, specify the types and concentrations of protease and phosphate inhibitors.

Response: Thank you for pointing this out. We have now included detailed information on the types and concentrations of protease and phosphatase inhibitors used in the Western blot experiments in the revised manuscript (Lines 380-382).

5. Line 513, please check the equation. It seems wrong, unless you use negative Hill coefficients.

Response: Thank you for pointing this out. We have corrected the equation and updated it in the revised manuscript (Line 668).

6. Line 570, ref. Chen et al is incomplete.

Response: Thank you for noting this, fixed.

7. Indicate in all the protocols also the pH between acidic stimulations.

Response: Thank you. In the revised manuscript, we have now included the pH conditions used between acidic stimulations in all relevant experimental protocols for clarity.

8. For the western blots in Fig. S8, a control from non-transfected cells needs to be included.

Response: Thank you for your suggestion. In the revised manuscript, we have included a control using non-transfected cells in the Western blot analysis, as shown in the updated Appendix Fig. S11. We also performed immunofluorescence staining on transfected cells to further validate the specificity of the antibody.

The PDB validation reports have been uploaded as source data for Figure 3

Dear Dr. Wang

Thank you for the submission of your revised manuscript to EMBO reports. We have now received the full set of referee reports that is copied below.

As you will see, all referees are very positive about the study and request only minor changes to clarify text and figures.

From the editorial side, there are also a few things that we need before we can proceed with the official acceptance of your study.

- Your manuscript will be published in our Reports section, which would require a combined Results & Discussion section.
- The funding information in the Acknowledgments and in the online manuscript tracking system must be congruent. Please add the following funds in the system: the National Key Research and Development Program of Heilongjiang province (2024ZXDXC28); the Longjiang Science and Technology Talents Spring Goose Support Program (CYQN24058); Heilongjiang Province (YQ2022H001) .
- Regarding the Author Contributions, we now use CRediT to specify the contributions of each author in the journal submission system. Therefore, please remove the Author Contributions from the manuscript file and make sure that the author contributions in our online manuscript tracking system are correct and up-to-date. The information you specified in the system will be automatically retrieved and typeset into the article. You can enter additional information in the free text box provided, if you wish.
- Please add a callout for Fig. 3E in the text.
- Reagent and Tools Table: Please remove the Instructions paragraph from the file.
- Data availability section: You have provided a specific URL for 9JF7 but not for EMD-61429. Could you please add the latter? Could you also add a specific URL for PQ282429?
This section should only refer to data deposited to external repositories and not to other other data. The address for plasmid requests could remain in this section, however.
- The synopsis image looks really great, but when I reduce its size to the final one (550 pixels width) it appears a bit blurry. Could you maybe check again? We need the final image at 550 pixel width. I know that's quite small, so please make sure that all text is readable and that the image quality is adequate at this resolution. Thank you.
- Please zip the legends as README.txt files with each movie and then upload the zipped folders.
- Please provide a higher resolution appendix file as the images appear pixelated under filters.
- Figure Legends (main + EV):
Please provide the exact p values in the legends of figures 1K, 2H, 3G, 4C.
Please note that the error bars are not defined in the legend of figure 5F.
- Appendix Fig S8A would need a scale bar.
- "Materials and Methods" should be renamed to "Methods"
- The correct title of the disclosure paragraph is: "Disclosure and competing interests statement".
- Sections need to be named and the order should be corrected: Title page - Abstract - Keywords - Introduction - Results - Discussion - Methods - Data Availability - Acknowledgements - Disclosure and Competing Interests Statement - References - Figure Legends - Table(s) - Expanded View Figure Legends.

Please provide a point-by-point response to both, the referee reports and the editorial points. The latter will speed up our checks. Thank you.

With kind regards,

Martina

=====

Referee #1:

The authors have put in substantial efforts addressing review critiques by providing abundant new data and information. The manuscript is substantially improved as a result. Most of my previous concerns have been satisfactorily addressed; I appreciate the detailed explanations the authors kindly provided.

Regarding the physiological role of the OFF response (critique point #1), the authors offered an interesting argument based on the inverse correlation between liquid volume and acidification rate. This is helpful. In order to complete their argument, the authors are encouraged to plot the relationship between pH change rate and liquid droplet size shown in Figure 11 and extrapolate to the size range of the centipede antennae. This will demonstrate the likely effect of CO₂ level on the OFF current size.

Referee #2:

The authors have done extensive revisions and have addressed all my concerns.

Referee #3:

The authors have revised and improved the manuscript. They have addressed my comments. One point, mentioned as my previous remark 1.1, concerning the effect of calcium on the currents, requires a further change in the manuscript. Calcium was shown with ASICs and related channels as a key regulator of pH dependence and ion permeation. Therefore it is critical to address experimentally the role of calcium, not only for blocking the currents, with PDPNac1, for example by providing the figure shown in the response to the reviewers. Maybe I have missed it, but I did not find this data in the revised manuscript. And one small typo, Figure S11, "western bolt" should be "western blot".

Response to Editor

- Your manuscript will be published in our Reports section, which would require a combined Results & Discussion section.

Response:

Thank you for your guidance. We have now combined the Results and Discussion sections and adjusted the content accordingly to improve the flow and coherence of the manuscript. Importantly, these changes do not alter the conclusions or the data presented.

- The funding information in the Acknowledgments and in the online manuscript tracking system must be congruent. Please add the following funds in the system: the National Key Research and Development Program of Heilongjiang province (2024ZXDXC28); the Longjiang Science and Technology Talents Spring Goose Support Program (CYQN24058); Heilongjiang Province (YQ2022H001).

Response:

We have added the provincial funding agencies in the submission system as requested, and they are also acknowledged in the revised manuscript.

- Regarding the Author Contributions, we now use CRediT to specify the contributions of each author in the journal submission system. Therefore, please remove the Author Contributions from the manuscript file and make sure that the author contributions in our online manuscript tracking system are correct and up-to-date. The information you specified in the system will be automatically retrieved and typeset into the article. You can enter additional information in the free text box provided, if you wish.

Response:

We have removed the Author Contributions from the manuscript file.

- Please add a callout for Fig. 3E in the text.

Response:

We have added the callout for Fig. 3E in the text (line: 208).

- Reagent and Tools Table: Please remove the Instructions paragraph from the file.

Response:

We have removed the Instructions paragraph from the Reagent and Tools Table.

- Data availability section: You have provided a specific URL for 9JF7 but not for EMD-61429. Could you please add the latter?

Could you also add a specific URL for PQ282429?

This section should only refer to data deposited to external repositories and not to other other data. The address for plasmid requests could remain in this section, however.

Response:

The URL for EMD-61429 has been provided (line: 696), and we have also added the GenBank URL in the manuscript (line: 694). Although it is not accessible at this stage, it will become valid upon article publication, with the accession number available for reference.

- The synopsis image looks really great, but when I reduce its size to the final one (550 pixels width) it appears a bit blurry. Could you maybe check again? We need the final image at 550 pixel width. I know that's quite small, so please make sure that all text is readable and that the image quality is adequate at this resolution. Thank you.

Response:

Thank you for your reminder. We have checked the image again and uploaded both .tif and new .jpg files in the system. Please contact us if there are any further issues.

- Please zip the legends as README.txt files with each movie and then upload the zipped folders.

Response:

We have prepared a zip of README.txt file for the legends of each movie.

- Please provide a higher resolution appendix file as the images appear pixelated under filters.

Response:

We have updated the appendix file with higher-resolution images. Please check the revised version, and do not hesitate to contact us if any images still appear unclear.

- Figure Legends (main + EV):

Please provide the exact p values in the legends of figures 1K, 2H, 3G, 4C.

Please note that the error bars are not defined in the legend of figure 5F.

Response:

As requested, we have provided the error bars and exact p values in both the figure legends and the Appendix files.

- Appendix Fig S8A would need a scale bar.

Response:

Thank you for your reminder, we have provided the scale bar, updated in Appendix Fig. S10A.

- "Materials and Methods" should be renamed to "Methods"

Response:

Renamed as requested.

- The correct title of the disclosure paragraph is: "Disclosure and competing interests statement".

Response:

Renamed as requested.

- Sections need to be named and the order should be corrected: Title page - Abstract - Keywords - Introduction - Results - Discussion - Methods - Data Availability - Acknowledgements - Disclosure and Competing Interests Statement - References - Figure Legends - Table(s) - Expanded View Figure Legends.

Response:

Reordered as requested.

Response to Referees

Referee #1:

The authors have put in substantial efforts addressing review critiques by providing abundant new data and information. The manuscript is substantially improved as a result. Most of my previous concerns have been satisfactorily addressed; I appreciate the detailed explanations the authors kindly provided.

Regarding the physiological role of the OFF response (critique point #1), the authors offered an interesting argument based on the inverse correlation between liquid volume and acidification rate. This is helpful. In order to complete their argument, the authors are encouraged to plot the relationship between pH change rate and liquid droplet size shown in Figure 1I and extrapolate to the size range of the centipede antennae. This will demonstrate the likely effect of CO₂ level on the OFF current size.

Response:

We sincerely appreciate the reviewer's positive assessment of our revised manuscript and their recognition of the substantial efforts we have invested in improving the study. We are also grateful for this insightful suggestion. The statistical analysis of droplet diameter vs. pH transition time has been added to the Appendix Fig. S3, and the results are described in the manuscript (Line: 108-109).

Referee #2:

The authors have done extensive revisions and have addressed all my concerns.

Response:

We thank the referee for their positive feedback and are pleased that our revisions have addressed their concerns.

Referee #3:

The authors have revised and improved the manuscript. They have addressed my comments. One point, mentioned as my previous remark 1.1, concerning the effect of calcium on the currents, requires a further change in the manuscript. Calcium was shown with ASICs and related channels as a key regulator of pH dependence and ion permeation. Therefore, it is critical to address experimentally the role of calcium, not only for blocking the currents, with PDPNaCl, for example by providing the figure shown in the response to the reviewers. Maybe I have missed it, but I did not find this data in the revised manuscript.

And one small typo, Figure S11, "western bolt" should be "western blot".

Response:

We sincerely thank the reviewer for the careful re-evaluation of our revised manuscript and for the valuable comments. In response to the concern regarding the effect of calcium on the currents, we have included new experimental data demonstrating that calcium modulates the pH dependence of PDPNaCl. Additional data further show that calcium ions decrease the pH₅₀ of PDPNaCl, consistent with the well-established regulatory role of calcium in ASICs. These results suggest that the modulatory function of calcium may be evolutionarily conserved within the DEG/ENaC family. The new data have been incorporated into the Appendix Fig. S4 and are described in the Results and Discussion section (line: 133-139).

We also thank the reviewer for pointing out the typo in Appendix Figure S11 (now updated to Fig. S13). This has been corrected from "western bolt" to "western blot" in the revised version.

Dr. Yunfei Wang
Northeast Forestry University
College of Wildlife and Protected Area
26 Hexing Road
Harbin, Heilongjiang 150040
China

Dear Dr. Wang,

I am very pleased to accept your manuscript for publication in the next available issue of EMBO reports. Thank you for your contribution to our journal.

Yours sincerely,
